# Short- and long-read metagenomics expand individualized structural variations in gut microbiomes

Liang Chen[1,9], Na Zhao[1,9], Jiabao Cao[1,2,9], Xiaolin Liu [1,2,9], Jiayue Xu[1,9], Yue Ma [1], Ying Yu[1,2], Xuan Zhang[1], Wenhui Zhang[1], Xiangyu Guan [1], Xiaotong Yu[3], Zhipeng Liu[4], Yanqun Fan[4], Yang Wang[5], Fan Liang [5], Depeng Wang[5], Linhua Zhao[3], Moshi Song [2,6,7,8✉] & Jun Wang [1,2✉]

In-depth profiling of genetic variations in the gut microbiome is highly desired for understanding its functionality and impacts on host health and disease. Here, by harnessing the long read advantage provided by Oxford Nanopore Technology (ONT), we characterize fine-scale genetic variations of structural variations (SVs) in hundreds of gut microbiomes from healthy humans. ONT long reads dramatically improve the quality of metagenomic assemblies, enable reliable detection of a large, expanded set of structural variation types (notably including large insertions and inversions). We find SVs are highly distinct between individuals and stable within an individual, representing gut microbiome fingerprints that shape strain-level differentiations in function within species, complicating the associations to metabolites and host phenotypes such as blood glucose. In summary, our study strongly emphasizes that incorporating ONT reads into metagenomic analyses expands the detection scope of genetic variations, enables profiling strain-level variations in gut microbiome, and their intricate correlations with metabolome.

[1] CAS Key Laboratory of Pathogenic Microbiology and Immunology, Institute of Microbiology, Chinese Academy of Sciences, Beijing, China. [2] University of Chinese Academy of Sciences, Beijing, China. [3] Guang'anmen Hospital, China Academy of Chinese Medical Sciences, Beijing, China. [4] Biotree-Shanghai, Shanghai, China. [5] GrandOmics Biosciences, Beijing, China. [6] State Key Laboratory of Membrane Biology, Institute of Zoology, Chinese Academy of Sciences, Beijing, China. [7] Institute for Stem Cell and Regeneration, Chinese Academy of Sciences, Beijing, China. [8] Beijing Institute for Stem Cell and Regenerative Medicine, 100101 Beijing, China. [9] These authors contributed equally: Liang Chen, Na Zhao, Jiabao Cao, Xiaolin Liu, Jiayue Xu. ✉email: songmoshi@ioz.ac.cn; junwang@im.ac.cn

The human gut microbiome contributes to host metabolic and immune homeostasis[1], and microbial dysbiosis has been shown to underly a wide range of diseases including metabolic and immune disorders, central nervous system pathologies, and cancer[2]. Methodologically, most compositional and functional insights about the microbiome have been obtained based on shot-gun metagenomic sequencing data[3], which has supported dedicated profiling of microbiomes in terms of single-nucleotide polymorphisms (SNPs)[4] and structural variations (SVs)[5,6] for various populations. Such profiling has revealed highly distinct yet temporally stable genetic variations at the individual level, which have been conceptualized as "microbiome fingerprints". Importantly, beyond straightforward comparisons based on sole taxonomical abundances, the additional layers of genetic variations in an individual's microbiome fingerprint have been linked to microbial metabolism and consequently to host health.

Recent advances in sequencing technologies such as Oxford Nanopore Technology (ONT) provide unique opportunities for investigating gut microbiome variations and functionality. The relatively longer read length with ONT has already been widely utilized for assembling complex eukaryotic genomes[7] and for resolving difficult regions including tandem repeats and large structural variations;[8] in microbiome studies ONT data support both improved metagenomic assemblies and functional annotations[9], and have facilitated the studies of the transmission of antibiotic-resistant genes in various settings[10]. Incorporating ONT long reads perceptibly increases confidence in microbiome fingerprinting, as improved metagenomic assemblies result in more complete bacterial and viral genomes[11]. Notably, sufficiently long reads are also capable of covering a large range of genomic regions including diverse structural variations, thus enabling direct read-level validations of fingerprints[12].

Here, we present the assembly of hundreds of gut microbiomes from healthy humans using an approach that combines ONT and Illumina (short) read data (Fig. 1a). At the population level, we expanded the personalized signatures of gut microbiome to include a wider range of SVs (insertions and inversions, besides deletions), in which we demonstrated the personalized microbial fingerprints confering strain-level differentiations with respect to metabolic functions and host blood glucose.

## Results

**Hybrid sequencing improves the quality of human gut metagenome assembly.** We first established a hybrid pipeline that incorporated both ONT and Illumina reads, and enabled both metagenome assembly and consequent data analysis. Using ONT and Illumina reads generated from ZymoBIOMICS™ Microbial Community containing eight strains of bacterial DNAs with equal molars, the pipeline achieved high completeness (94.54–99.75%), a low contamination rate (0–6.97%), and >97.6% average nucleotide identity (ANI) for re-constructed bacterial genomes (Methods, Supplementary Fig. 1a–e). By contrast, using only Illumina reads led to much shorter contigs, with no detectable changes in the numbers of binned genomes and their completeness; analyzing ONT reads with methods aimed at generating circularized genomes, the genome completeness was overall below 93%, combined with a reduced number of binned genomes (Supplementary Fig. 1b and Supplementary Data 1). We further examined whether ONT reads introduced more errors into contigs and reduced open-reading-frames (ORFs) numbers in a genome (i.e., coding density), and found that hybrid assemblies had no detectable reduction in coding density compared to that Illumina-only assemblies, while using only ONT reads led to a significant decrease in coding density (Supplementary Fig. 1c).

We then applied our hybrid assembly strategy to two cohorts of human gut microbiome data: a cross-sectional cohort of 100 healthy individuals (Supplementary Data 2) and a time-series cohort comprising ten healthy individuals (Supplementary Data 3), each with ten fecal samples collected continually. The results showed that our hybrid assembly approach greatly improved the quality of the metagenomic contigs for the 200 fecal samples from the two cohorts. Using on average $1.4 \times 10^6$ ONT reads (mean length of 5683 bp) and $5.6 \times 10^7$ Illumina 150-bp pair-end reads per sample, our pipeline assembled on average $7.1 \times 10^7$ contigs totaling $7.6 \times 10^{10}$ bp (Supplementary Fig. 1f and Table 1). Overall, hybrid assemblies had 17.3% fewer contigs and 5.1% more total assembled sequences as compared to the assemblies obtained using Illumina reads alone ($8.5 \times 10^7$ contigs and $7.2 \times 10^{10}$ bp, respectively, Table 1). Of note, the average N50 value more than tripled for the hybrid assemblies (9283 bp) compared to the short read alone assemblies (2962 bp).

We then binned the contigs obtained from the hybrid assembly into metagenome-assembled genomes (MAGs) representing individual bacterial species, resulting in a total of 9612 MAGs (20–83 MAGs per sample) with an average N50 of 117 kb; there remained 692 MAGs after the removal of redundant MAGs (i.e., belonging to the same bacterial species, Fig. 1b, c and Table 1). Among those, 623 corresponded to the available genomic bins in Unified Human Gastrointestinal Genome (UHGG) database, and 208 among them had higher quality in our hybrid assembled results; the rest of a total of 67 genomic bins were novel genomes. There were two MAGs fewer after dereplication, as a few relatively close MAGs in our cohorts and UHGG collection that were previously not clustered together (with dRep[13] v2.2.4) are now clustered with a higher version of dRep (we used v2.6.2). Regarding comprehensiveness, 159 (22.97%) of the non-redundant MAGs included all three types of rRNA sequences (23S, 16S, and 5S), and 448 (64.74%) of the MAGs had at least one type of rRNA. By contrast, Illumina-only assembly produced 11% fewer non-redundant MAGs (616) with roughly half the average N50 value (65 kb), among which only 9 (1.46%) MAGs had all three types of rRNA sequences and only 258 (41.88%) MAGs had at least one type. Across all the samples, the most commonly occurring MAG was for *Fusicatenibacter saccharivorans* (present in 172 samples) followed by *Anaerostipes hadrus* (150) and *Agathobacter rectalis* (148), and there were 189 species present as MAGs in >10 samples (Supplementary Data 4 and Supplementary Fig. 2).

**Expanding the scope of detected structural variations in gut microbiome.** The improved metagenomic assemblies with N50 > 110 kb provided an opportunity to expand the investigative scope for genetic variations in human gut microbiome. In particular, considering that deletions remain the major type of structural variations studied in gut microbiomes based on mapping against reference genomes[5,6], while insertions and inversions— and especially the large ones that cannot be covered with short reads—remain elusive (both discovery and validation). The longer length of the ONT reads improved the metagenomic assemblies and enabled the discovery of expanded SVs including insertions/inversions, while simultaneously providing the opportunity to conduct direct validations for large SVs at the read level across the cohort.

In our cross-sectional cohort, we reliably detected multiple types of structural variations by comparing MAGs. For each of the 189 bacterial species present in >10 individuals, we used the MAG with highest scores evaluated by dRep v2.6.2[13] as the reference for comparing the same-species MAGs from other samples. This identified a total of 317,558 insertions, 342,129 deletions, and 1373 inversions (Fig. 1d). Notably, SVs larger than

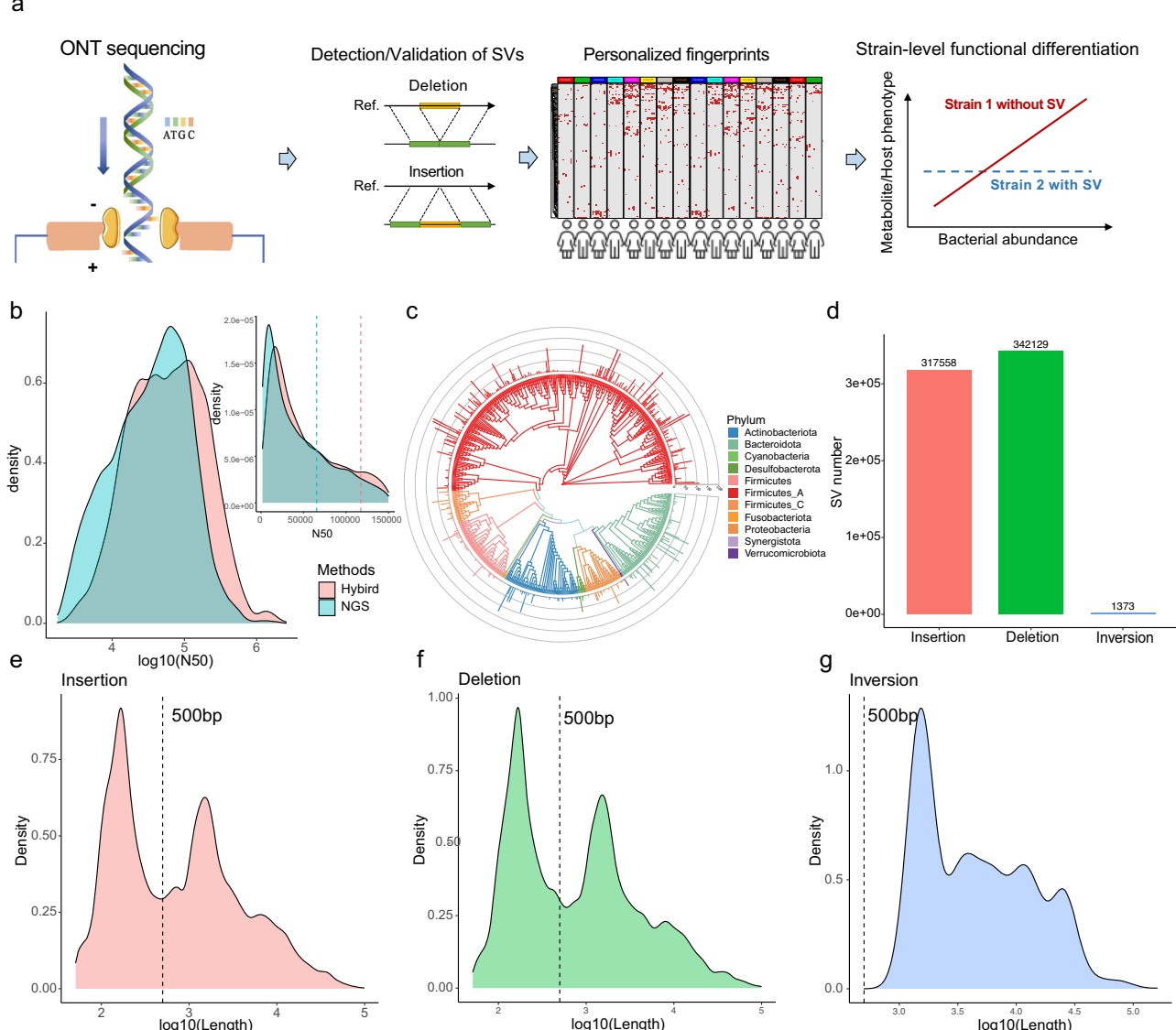

**Fig. 1 ONT reads improved metagenomic assembly, empowered structural variations (SVs) detection and validations. a** Schematic representation of workflow of this study. Top, utilizing the long reads from ONT we improved metagenomic assemblies in hundreds of gut microbiome, enabled detection of large SVs and notably including insertions and inversions, which are highly personalized gut microbial signatures and complicate the correlations to metabolites or host health indicators. **b** Distribution of contig lengths for Illumina-only approach (NGS) and hybrid assembly (Hybrid), the full distribution is shown in log-scale in the main graph; and part of the detailed distribution is shown in upper right panel including dash lines showing the mean N50 values for binned metagenome-assembled genomes (MAGs) in Illumina-only approach (NGS) and hybrid (Hybrid) assembly. **c** Phylogenetic distribution of 692 MAGs binned from 200 gut microbiome samples using hybrid assembly. Colors denote major phyla of gut microbiome and the lines in the outer circle indicate number of occurrences for that MAGs (species) in the 200 samples. **d** Total number of insertions, deletions, and inversions discovered for the 189 MAGs (species) with >10 occurrences in our cohort, with one representative MAGs for each MAGs (species) and rest of MAGs compared to that representative (see Methods and Results). **e–g** show the length distribution of discovered insertions, deletions, and inversions, in particular, large SVs (> 500 bp) accounting for ca. 50% of insertions and deletions, and all inversions.

**Table 1 Summary of key parameters for hybrid assembly and Illumina-only assembly per sample.**

|  | Hybrid assembly | Illumina-only |
| --- | --- | --- |
| Raw reads | 1.47E6 (3.97E5−9.23E6)[a] | 5.64E7 (3.41E7−1.07E8) |
| Total nucleotide of raw reads | 6.08E9 (1.72E9−2.08E10)[a] | 8.46E9 (5.12E9−1.61E10) |
| Contigs | 3.55E5 (1.12E5−7.77E5) | 4.29E5 (1.35E5−8.63E5) |
| Total nucleotide of contigs | 3.82E8 (1.69E8−7.05E8) | 3.64E8 (1.60E8−6.63E8) |
| Contig N50 | 9283 (2002-31,606) | 2962 (745-11,932) |
| MAGs numbers | 50 (21-196) | 44 (14-74) |

For each parameter, the average value and range are shown.
[a]Denotes the number for ONT reads used in hybrid assembly.

>500 bp comprised a large proportion for each SV type, including 170,329 (53.63%) insertions, 184,037 (53.80%) deletions, and all of the 1373 inversions (Fig. 1e–g). Interestingly, we observed two peaks in the distributions of insertion and deletion, for which we hypothesized that the two peaks of SVs were results of different biological processes in prokaryotic genome, especially with regard to transposon/prophage and other mobile elements' activities. Thus, we analyzed randomly selected SVs within two peaks (within 140–160 bp and 1050–1150 bp, respectively), and predicted the prophage and extrachromosomal mobile genetic elements (eMGEs) using blastn based on the mMGE database[14]. Results indicated significant differences between SVs within of the two peaks and mobile elements are significantly higher in short SVs: prohages in short vs long SVs: deletion ($p = 2.82e-06$), insertion ($p = 2.93e-05$); and eMGEs in short SVs vs long SVs: deletion ($p = 4.385e-07$), insertion ($p = 0.0005129$, all with Wilcox test). We thus infer the short SV are more likely results of phage integration and other mobile elements compared to longer ones. Yet, as not all SVs have detectable mobile elements, this offers only a partial and plausible explanation; we presume that the other SVs are results of replication error or recombination events but mechanistic validations are not available from limited studies focusing on SVs in bacteria.

We evaluated the reliability of detected SVs by identifying ONT reads that could directly cover a SV and its flanking regions, based on re-mapping to either the reference MAG or to the MAG containing the SV. Manual inspections ultimately confirmed that >97% of a randomly selected set of SVs were supported by multiple ONT reads, thus confidently validating the existence of specific SVs with single-molecule reads covering the corresponding genomic regions (Fig. 2a and Supplementary Fig. 3); and the mapping results simultaneously indicated low heterogeneity in terms of SVs of same-specie bacterial genome within the same individual.

A clear trend in our SV dataset was that the frequencies of SVs in bacterial genomes were uneven among taxonomical groups. At the species (MAGs) level, our data showed that the total number of SVs discovered was proportional to the number of MAGs but also to genome size across all the samples. Accordingly, to account for uneven SV distributions, our findings emphasized that any comparison between different taxonomical groups should include averaging based on pair-wise comparisons and should correct for genome size (Supplementary Fig. 4). Such analysis of average SV numbers between a given MAG and reference MAG (standardized to per 1 Mb genome across taxonomical groups) revealed that at the phylum level, directly following the highly-diverse Firmicutes with median SVs of 20.4, the *Akkermensia*-containing bacterial phylum Verrucomicrobia had the second highest number of SVs (median 19.5, Fig. 2b, c), while the Desulfobacteroita and Proteobacteria phyla had the lowest numbers of SVs (median 8.6 and 11.5, respectively, Fig. 2b). Verrucomicrobia contains only one established species, *Akkermensia muciniphila*, and it is currently being widely studied for metabolism-modulating effects[15,16]. Here, their high inter-individual genetic diversity and consequently larger pan-genome called for special attention in resolving strain-level differences in translational studies[17].

**SVs as highly personalized signatures of gut microbiome are function-informative**. Analysis of our ONT-read-informed SV dataset strongly supported the idea that SVs can define informative, personalized gut microbiome signatures. We simultaneously detected high inter-personal variabilities in our cross-sectional cohort data yet low intra-personal and temporal variabilities in our time-series data. Analysis of the 189 MAGs used in

our SV discovery effort revealed that a median of 16.7 SVs per Mb genome were found between MAGs from different individuals, in contrast to a median of 0 SVs per MAG within a single individual along ten consecutive time points (Wilcox test $p < 2.2e-16$, Fig. 2d). Thus, SVs reliably distinguished bacterial species and collectively the gut microbiome between different individuals. It bears emphasis that our findings here mirrored the recent discovery from the LifeLines cohort that structural variations comprise "fingerprints" that can distinguish person-specific bacterial species. Furthermore, beyond the SV fingerprints reported from previous studies which have focused on deletions[5,6] (owing to the limits in short read-based assemblies), our dataset revealed that a nearly equal number of insertions were present alongside such deletions. The time-resolved samples of our dataset also enabled us to complement these findings about person-specific gut microbiome signatures: within *ca*. 10 days (Fig. 2d and Supplementary Fig. 5), the genome structure of the same species remained stable indicating that the strain differentiation/replacement observed over three years in the LifeLines cohort[6] could be results of gradual SV accumulations.

As SVs in genome cause breaking points in genome and consequently might affect the functionality of genes[18], we investigated the functional distribution of genes in reference MAGs that contain such breaking points. As we observed relatively high inter-individual variations in terms of number and types of SVs compared to the reference MAGs, addressing each individual and bacterial genomes in terms of SV function is difficult; also the relatively low number of SVs in each individual and per MAG (16.7 per Mb) prevented informative enrichment analysis, thus we carried out functional enrichment analysis of SV-related gene functions at a population scale. Using KEGG pathway and against the baseline for all the genes predicted in the reference MAGs, enrichment analysis revealed a total of 267 enriched pathways for the insertions and the deletions (Fig. 3a and Supplementary Data 5); no pathways were significantly enriched for the inversions, likely owing to their smaller number than insertions/deletions. There were 19 metabolism-related pathways among the top 30 most affected pathways (ranked based on the extent of enrichment), including for example "glycan degradation", "sphingolipid metabolism", and those for the metabolism of diverse carbohydrates (Fig. 3a and Supplementary Data 5). There was also enrichment for pathways associated with processing of environmental information, including the phosphotransferase system (PTS), ABC transporters, and two-system transporting systems, genes of which have been implicated in bacterial toxin production and in conferring antibiotic resistance (Fig. 3a and Supplementary Data 5). Interestingly, the enrichment results agreed with the findings in a previous report studying SV-affected genes in Israeli and Dutch populations[5,6], indicating potentially the universal characteristics in terms of human gut microbial SV-affected genes.

**SVs complicate bacterial associations to metabolites and host phenotypes**. To investigate the functional consequences of SVs, in particular to the metabolism of microbiome, we carried out metabolome analysis in the fecal, serum and urine samples in the cross-sectional cohort. Analysis based on the metabolome of different samples in the cross-sectional cohort showed that SVs complicated the correlations between bacterial species and metabolites, leading to strain-level functional differences within the same species of bacteria significantly correlated with metabolites. Namely, SVs led to potential disruption of gene function and in the subgroup containing SVs abolished significant correlations between bacterial abundance and metabolites; by contrast, the subgroup without SVs maintained significant correlations. In

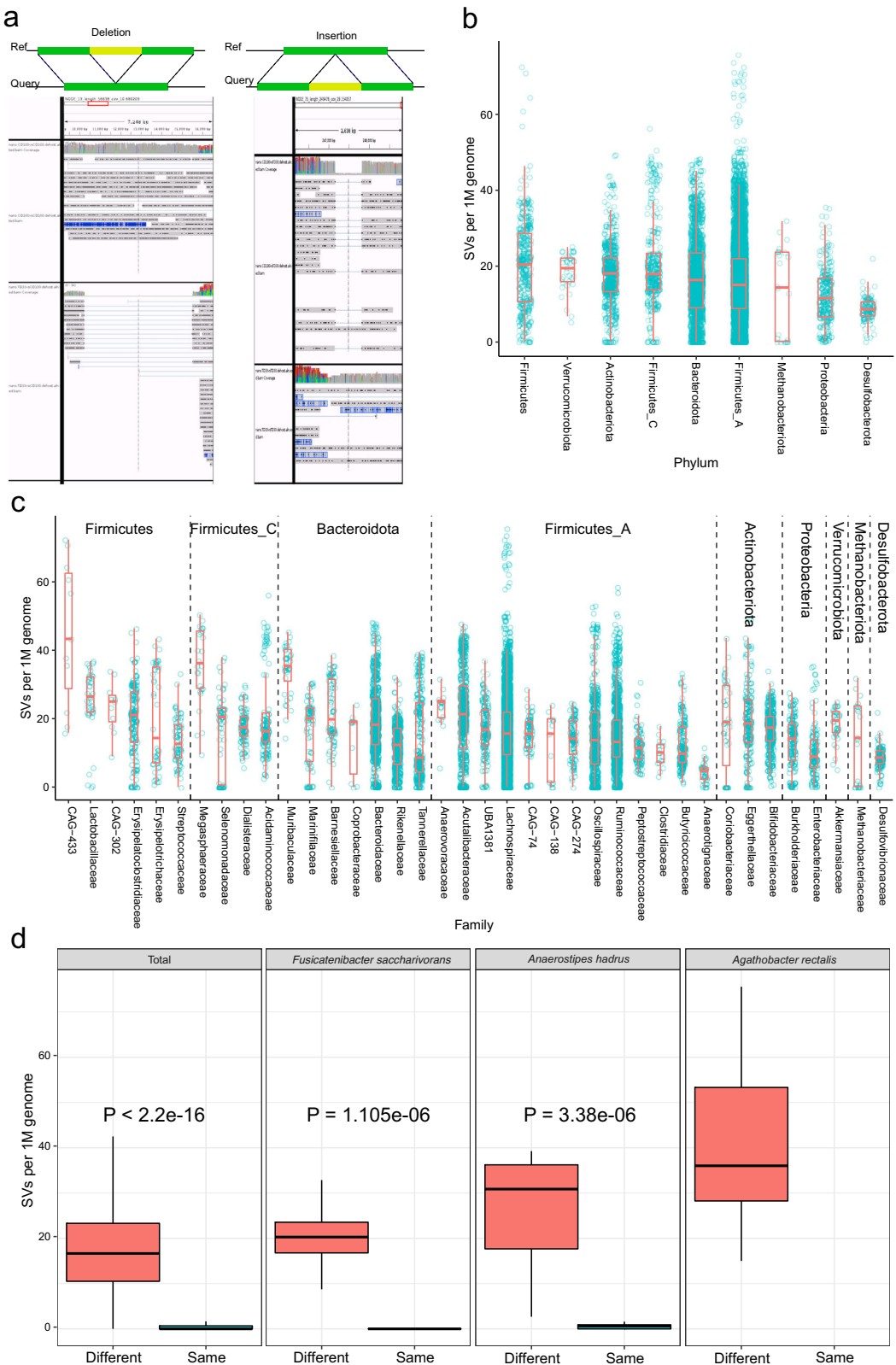

this finer-scale analysis we found among 11 bacterial species with significant correlations (FDR < 0.1) to metabolites in the fecal, serum, or urine metabolome, a total of 889 SV-affected genes complicated bacterial-metabolite correlations (Fig. 3b, c, Supplementary Fig. 6, and Supplementary Data 6).

We discovered that in detail, 753 pairs involved 70 SVs that correlated with 74 fecal metabolites (out of 458), 134 pairs

involved 31 SVs that correlated with 66 urine metabolites (out of 396), and 2 pairs involving 2 SVs with 2 serum metabolites. Among these results, our discovery and inclusion of insertions revealed that the expansion of SVs increased the power of discovering SVs that could complicate bacterial-metabolite correlations. For instance, the previous discovery in Israeli population found that the correlation between inositol

**Fig. 2 Validation and characterization of structural variations (SVs) in human gut microbiome. a** Schematic representation of direct validations of structural variations (SVs) using long ONT reads. Using the upper metagenome-assembled genome (MAG) as reference, deletions (left) and insertions (right) were identified in the lower MAG (belonging to the same bacterial species, from a different sample), and mapping long reads from different sample against representative sequences resulted in reads directly covering deletion (left) or insertion (right) and flanking regions, thus validating the presence of these SVs at read-level. **b**, **c** Phylum-level and family-level distributions in the number of major types of SVs (insertions and deletions) across different taxonomical groups, corrected for corresponding genome size (SVs per 1 M genome). High variability of SV numbers can be found among different phyla and bacterial families (see Results). **d** Comparison of average number of SVs per 1 Mb genome between all of 189 MAGs used for SV detections, as well as individual MAGs from the three most common same species, between different individuals in the cross-sectional cohort (left of each boxplot) and from different samples within the same individual in the time-series data (right of each boxplot). In all four cases, inter-individual SVs numbers are significantly higher than that of intra-individuals (two-sided Wilcoxon test, $n = 4093, 83, 91, 63$ separately, all $P < 2e-10$), suggesting SVs can be used as fingerprints in human gut microbiome to distinguish different individuals. Data are presented as box plots with whiskers at the 5th and 95th percentiles, the central line at the 50th percentile, and the ends of the box at the 25th and 75th percentiles.

concentrations and *A. hadrus* was confounded by a deletion in bacterial genome. By contrast, our study found that both insertions and deletions occurring at a gene locus (annotated to be K02014, iron complex outer-membrane receptor protein) in *Bacteroides uniformis* genome led to the loss of significant correlations between relative abundance of this bacteria species and inositol concentrations in urine (Supplementary Fig. 7).

Among the SV-confounded associations between bacterial species and metabolites, we found that *A rectalis* was significantly associated with fecal fructose-1-phosphate (F1P) when ignoring SVs (Spearman's $\rho = 0.28$, $p = 0.0053$, FDR = 0.035, Fig. 3e), among other bacteria. Further analysis accounting for the status of 12 SV-affected genes complicated the correlations between *F. saccharivorans* and the concentrations of neotrehalose in fecal samples (Fig. 3d, and Supplementary Data 6). Similarly, 33 SV-affected genes showed that the subgroup (strain) of bacteria containing SVs no longer had significant correlations with F1P (eg. K01193 in *A. rectalis*, $\rho = 0.18$, $p = 0.2$, FDR = 0.56, Fig. 3e). Among the metabolites and SV-affected genes, we found four metabolites affected by SVs and a total of 11 genes affected by SVs were mapped to four KEGG pathways, in which the SV-affected genes and metabolites were both involved, strongly suggesting the roles of SVs in shaping bacterial-metabolite correlations by affecting the function of relevant genes (Supplementary Data 7 and Supplementary Data 8). For instance, SV-affected genes confounding bacterial associations to neotrehalose included K01208 (cyclomaltodextrinase) and K05349 (beta-glucosidase), both belonging to the KEGG pathway "Starch and sucrose metabolism" (map 00500) (Supplementary Fig. 8 and Supplementary Data 7). Altogether, our data suggest that SVs are capable of introducing strain-level differences in metabolic functionalities in gut microbial species, and complicate the correlations between bacterial abundances and metabolites.

The complicated effects of SVs extended to that of the correlations between bacterial and host phenotypes. The two aforementioned fecal metabolites, F1P and neotrehalose, had significant negative correlations with the fasting blood glucose levels for the individuals in the cross-sectional cohort (Fig. 3f, g). The abundance of *F. saccharivorans* was significantly correlated with blood glucose (Spearman's $\rho = -0.38$, $p = 1e-4$, FDR = 2e-4), and among the individuals, the presence of SV at the locus (annotated to be K03655, putatively encoding an ATP-dependent DNA helicase RecG) defined a subgroup/strain of *F. saccharivorans* that was not significantly associated with blood glucose (Fig. 3h); and SVs at K01193 in *A. rectalis* also lead to decreased coefficients of association between bacterial abundance and glucose, although in the SV0 group the association was weaker ($p = 0.05$), potentially a result of lower sample size in subgroups (Fig. 3i). It has been reported that F1P can competitively inhibit the liver phosphorylase, which metabolizes glycogen to glucose, and thus fecal F1P potentially contributes to lowering blood

glucose;[19,20] for trehalose however, it is not yet clear the relevance to blood glucose. Our findings thus demonstrate that incorporating SVs increase the detection power in correlational analysis of bacterial and host health phenotypes, by controlling the effects of SVs that complicate the correlations between bacterial abundances and metabolite concentrations.

**Highly correlated prophage and CRISPR structures at the community level.** Incorporation of phages into bacterial genomes (forming prophages) and excision of existing prophages may both introduce SVs, and our improved metagenome from hybrid assembly facilitated the identification of prophages. Using the machine-learning-based tool ProphageHunter[21] with the 9612 MAGs (before redundancy removal) as the input, we identified a total of 2247 prophages, with genome sizes ranging between 1236 and 91,792 bp (Fig. 4a). Phylogenetic assignment based on concatenated capsid protein and terminase large subunits divided the prophages into two main viral families: *Siphoviridae* and *Myoviridae*. Furthermore, relying on long ONT reads we confirmed the direct linkage between prophage elements and flanking host bacterial genomes and established the associations between phage families and bacterial genera to include 1077 phage-host pairs (Fig. 4b). Among those, only 72 (6.69 %) were included in the current database of microbial-phage interactions, MVP[22]. By contrast, short read-based metagenomic profiling only discovered 1815 prophages, accounting for 80.77% in the hybrid assembly, thus showing that the ONT-improved metagenome facilitated prophage discovery.

Beyond prophages, microbial genomes also contained CRISPR-Cas systems for defense against re-infection by phages. It is now understood that the loci for these systems have spacers that record marker sequences for certain phages[23], thus leading to insertion/deletion variation within the same species. Among the same set of MAGs used in the aforementioned prophage analysis, we additionally discovered 150,058 CRISPR spacers with an average of $1665 \pm 560$ (mean $\pm$ SD) spacers per metagenomic sample, and average length of these spacers was $34 \pm 4.8$ nt (Supplementary Fig. 9). And the majority of spacers are not currently in reported databases, as only 17,600 or 11.73% were found in CRISPROpenDB[24] and 22,962 or 15.30% overlapped with spacers found in western populations gut microbiome[25] (Supplementary Fig. 10). Here, the improved metagenomic assembly again demonstrated the increased power of discovering particular genomic elements such as CRISPR spacers, whereas the same analysis in short read-based metagenomic assemblies revealed only 9542 spacers (6.36%, 15-fold lower), potentially due to the fact that short reads have difficulties in resolving highly identical repeat sequences between spacers.

The extensive diversity of prophages and CRISPR spacers helped define the informative, personalized SV fingerprints of

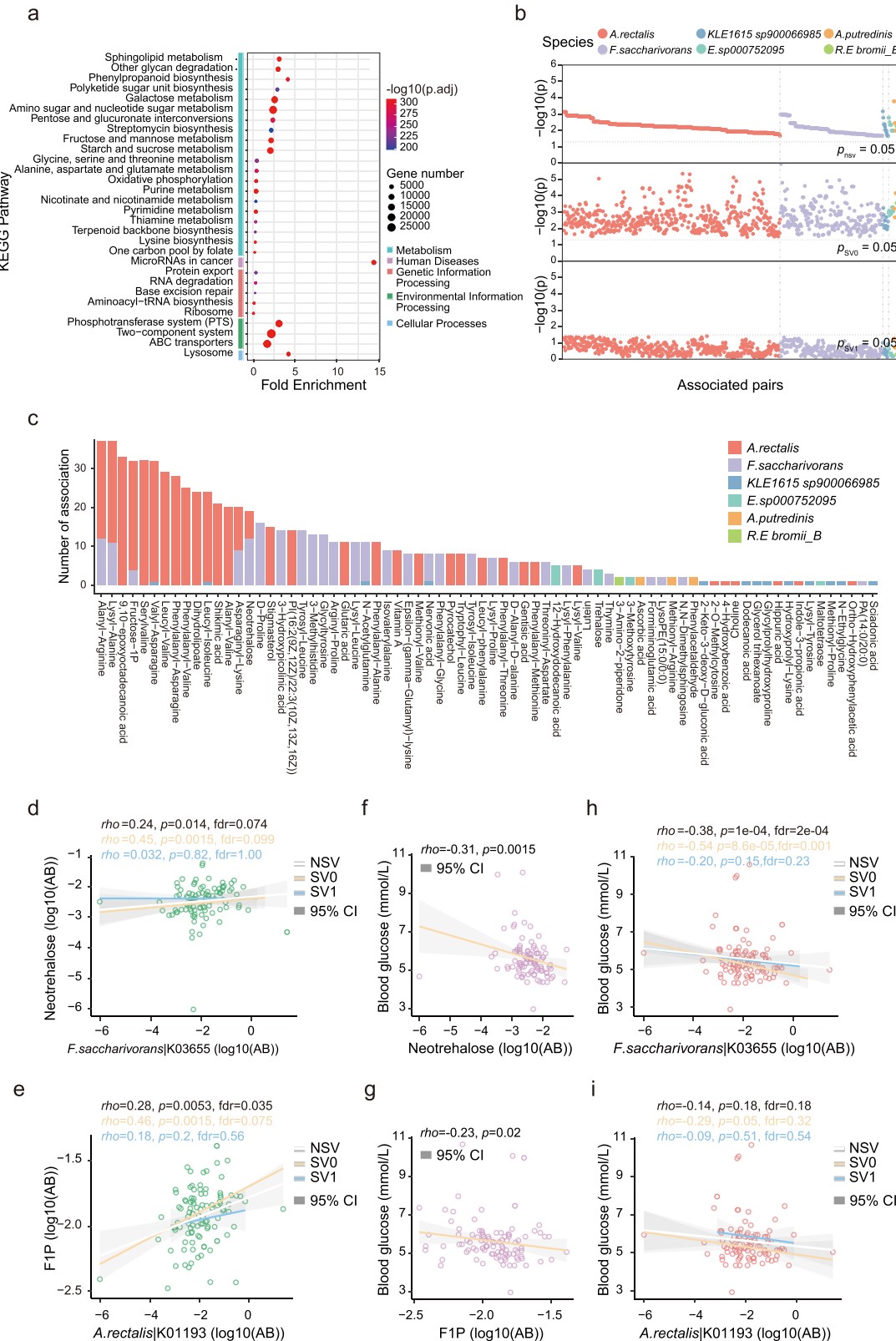

human gut microbiome. Indeed, we found that the inter-personal differences in our cross-sectional cohort were significantly higher than those within same individuals (in our time-series cohort), as measured by beta-distances based on prophage/CRISPR spacers (Jaccard distance calculated using prophage: Wilcoxon-test $p < 2e-16$; using CRISPR spacers: Wilcoxon-test $p < 2e-16$) (Supplementary Fig. 11). Further comparisons between the compositions of prophages and CRISPR spacers with respect to their community-level compositions revealed intriguing co-variations. Firstly, Procrustes analysis, which examines the correlations between different types of community differences, discovered significant correlations between their compositions across different individuals (Procrustes $r = 0.994$, $p < 0.001$, Fig. 4c) in the cross-sectional cohort. Secondly, further analysis identifying

**Fig. 3 Functional relevance of structural variations (SVs) in human gut microbiome. a** The top 30 categories in functional enrichment of SV-affected genes based on KEGG, metabolism-related pathways account for 19; $p$-values were from Fisher's test. **b** SVs influencing the gut bacteria-fecal metabolite correlations, upper panel indicates significant correlations (Spearman correlation, FDR < 0.1), the presence of SVs abolished significant correlations (lower panel, all $p$-values > 0.05) and the subgroup without SVs maintained significance (middle panel, all $p$-values < 0.05). Colors denote different bacterial species. **c** Overview of SV-affected bacteria-metabolite correlation pairs (Spearman). Colors and bars denote different bacteria and metabolites. **d** *Fusicatenibacter saccharivorans* was significantly correlated with neotrahlose (NSV, $n = 100$), SVs within gene K03655 lead to insignificant correlations (SV1, $n = 52$; $\rho$ (rho) = 0.032, $p = 0.82$), and different from the other subgroup/strain (SV0, $n = 48$; $\rho = 0.45$, $p = 0.0015$, FDR = 0.099). **e** Similar case for correlation between *A. rectalis* and Fructose-1-phosphate (F1P), SVs at K01193 leads to strain-level differences. **f, g** Fecal neotrahlose and F1P concentrations were significantly correlated with blood glucose ($n = 100$). **h** For *F. saccharivorans*, SVs within K03655 gene cause insignificant associations with blood glucose (SV1, $n = 52$; $\rho = -0.20$, $p = 0.15$), and the subgroup without SVs was significantly correlated with blood glucose (SV0, $n = 48$; $\rho = -0.54$, $p = 8.6e-5$, FDR = 0.001). **i** Similarly, for *A. rectalis*, SV within gene K01193 also leads to strain-level differences in correlation with blood glucose (SV0, $p = 0.05$, $n = 46$; SV1 group $n = 54$). The $\rho$ (rho) indicates the coefficient of spearman correlation, and $p$-values were adjusted with Benjamini-Hochberg procedures for FDR control. The shadings in (**d**–**i**) indicate the 95% confifence intervals (CI) and colors in **d**–**i** indicate different groups. NSV: all sample; SV0: subgroup without SV in bacterial gene; SV1: subgroup with presence of SV in bacterial gene. The details of subgroups are available in the Supplementary Data 6.

---

active phages from metagenomic reads revealed that only 47 out of 2247 identified prophages were potentially active (free-living), indicating a large reservoir of relatively inactive prophages incorporated in bacterial genome and constituted stable SVs.

## Discussion

Recent studies utilizing the developments in sequencing technology and analytical methods emphasize the ubiquity and functional importance of structural variations across humans and other animals, various plants, and more recently bacteria[3–6]. Structural variations are lower in occurrences in the genome than single-nucleotide polymorphisms (SNPs) but have higher chances of affecting gene functions as revealed in eukaryotic organisms[4–6]. However, profiling of SVs in microbes is still challenging, especially that short read- and mapping-based discovery of SVs is highly dependent on high-quality references and faces difficulties in identifying large insertions and inversions[5,6]. Incorporating long reads from ONT increases assembly quality and enables read-level validation of structural variations. For instance, in our large-scale sequencing of human gut microbiome we achieved metagenomic assemblies with more than tripled N50 values in contigs, and almost doubled, >100 kb of N50 values in genomic bins than those by only using short reads. With several studies have applied ONT sequencing to microbiome research, the major focus has been on improving assembly and required >200 Gb of reads per sample, an amount not yet feasible for population studies;[9] a cohort study with a focus on structural variations as well as associations with metabolic implications is first to be reported by our study. Compared to previous studies in human gut microbiome that mainly focus on deletions[5,6], our analysis additionally identified a large amount of insertions (nearly equal to the amount of deletions) and thousands of inversions, which greatly expanded the scope of detectable SVs. Furthermore, we demonstrated that large SVs could be more easily validated by ONT reads at the read level, increasing the confidence for future SV discovery and analysis in gut microbiome.

Our results revealed the heterogeneity in genome structural diversity among bacterial taxa groups, and re-emphasized the high diversity between individuals *versus* the high stability within the same individual, further supporting the notion that SVs could be used as "fingerprints" to distinguish the gut microbiome of different individuals[5]. Among the three studies available on gut microbiome structural variations, two were cross-sectional[6,26] and one study[5] compared samples collected three years apart, where they demonstrated that the SVs had significant changes within the same individual. In our study, the genome structure of the same species remained stable, but we acknowledge that our

study was unable to determine an appropriate time window to separate "short" (10 days) vs "long" (3 years) term to observe the occurrence of structural variations. In terms of potential functions, SVs in our study affected the integrity of the genes enriched in metabolic pathways for differentiated nutrient utilization, systems for transporting and environmental sensing, thus likely affecting or diversifying the metabolic capacities and subsequently enabling their occupation and competition of ecological niches in the same bacterial species. Combined with metabolome data collected in our study, we further established that SVs, acting as another layer of gut microbiome variations besides bacterial abundances that underlie the metabolic activities and metabolome, affected ca. 15% of the fecal and urine metabolites in our screen. It is thus likely that via modulating the associations between bacteria and metabolites such as neotrehalose and F1P, SVs confound the correlations of bacteria to metabolites and eventually to important host phenotypes (such as blood glucose), adding a layer of complexity in association between gut microbiome and host health. Yet, further functional experiments are warranted to establish the findings as our analysis are still limited to correlation inference. Our results add to the observed effects of microbiome SVs on metabolites and host phenotypes, consistent with the findings in the Israeli population and the Dutch LifeLines-DEEP cohort[5,6], all indicating the utility and importance of considering SVs in linking the microbiome to associated metabolomes from feces/serum, and eventually host health indicators.

Among the SVs, prophages and highly variable CRISPR elements comprised a significant proportion and our hybrid assembly strategy significantly improved the diversity of prophages and CRISPR spacers than short read-only approach did. The presence of prophages provided important information on the host range and specificity of phages, while CRISPR spacers recorded previous interactions with phages and novel CRISPR-Cas systems in gut microbiome could provide bases for new gene editing systems in the future[27]. We expanded the current knowledge of phage-bacterial host pairs by identifying >1000 new pairs of phage-host correlations. At the same time, nearly six-fold more of new CRISPR spacers were found in our data, indicating the still underappreciated diversity of CRISPR spacers, plus the high divergence between human gut microbiome from well-studied western populations and less frequently examined Asian populations[28].

To conclude, our population-scale microbiome analysis incorporating ONT reads simultaneously profiled multi-type, large structural variations in the human gut microbiome. SVs modulate bacterial functionality that impact host metabolome and health, calling for more finer-scale investigations of bacterial contribution to health and disease in humans, beyond a sole focus

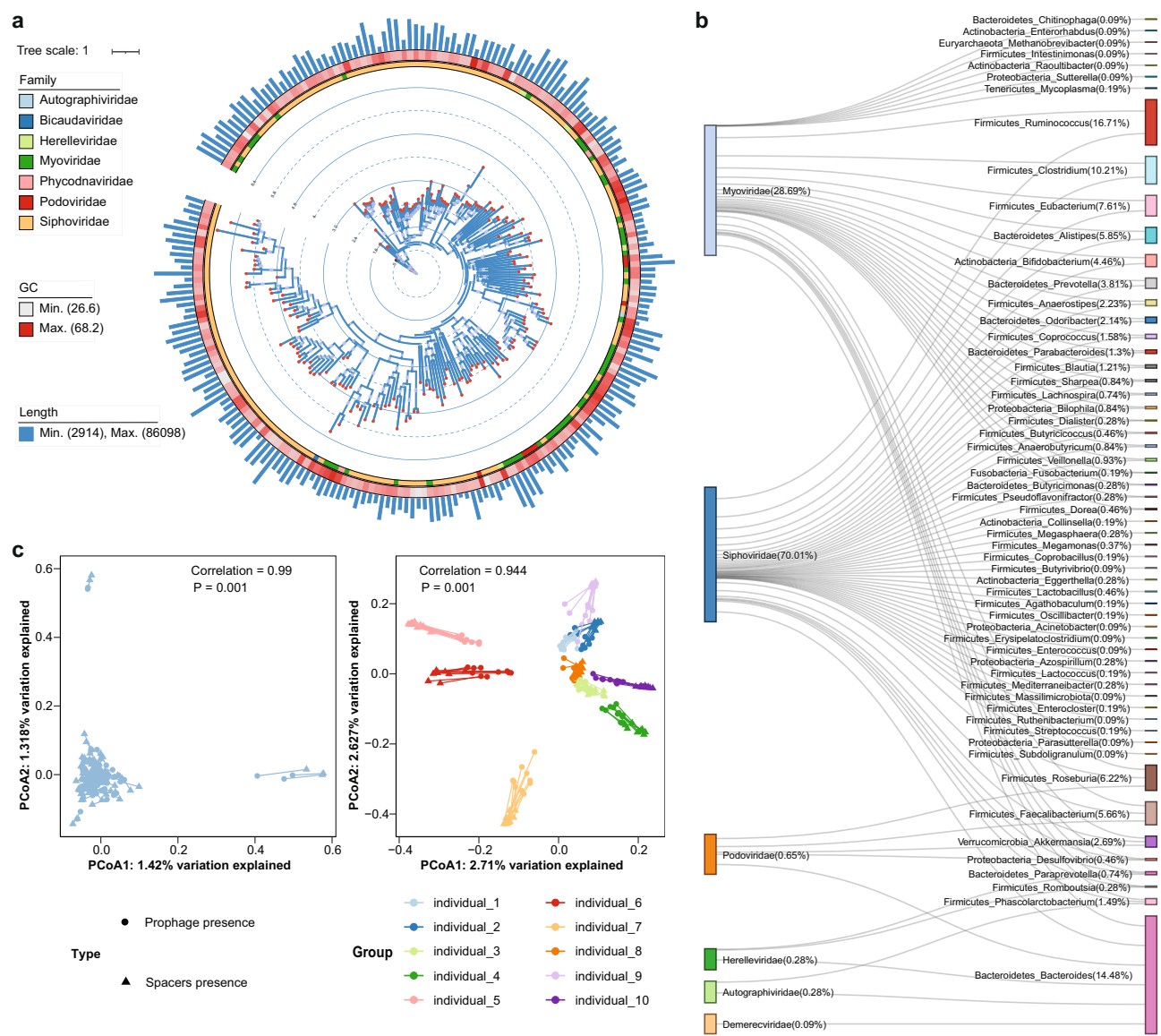

**Fig. 4 ONT-improved metagenome contained highly diverse prophages and CRISPR spacers in human gut microbiome. a** Phylogenetic distribution of 228 prophages with both complete major capsid protein (MCP) and terminase large subunit (TLS) proteins (see Methods and Results section) discovered from 200 samples, with length distribution from 2.9 to 86 kb (scaled in barplot). For each sequence, the assigned viral family was indicated in color and the length of each prophage was shown as bar length in the outer circle. **b** Prophage-host pairs determined by analyzing prophage sequences and flanking regions. Prophages were grouped at the family level and bacteria at the genus level, with each side showing the percentage of family/genus among all the sequences. **c** Procrustes analysis of prophage/CRISPR spacer structures in the cross-sectional cohort (left) and time-series cohort (right), showing significant correlations between their overall compositions and indicating the highly stable prophage/CRISPR spacers structures within the same individual across time points. The statistical significance of the procrustes results were assessed using function protest with 999 permutations. The $p$ value for both statistical tests was 0.001.

on bacterial abundance. Further incorporating ONT reads in gut microbiome research will enable in-depth dissection of time-specific gut microbiome functionality and deepen our understanding of various gut-disease axes in humans.

## Methods

**Establishing the hybrid assembly pipeline**. We used genomic DNA from ZymoBIOMICS™ Microbial Community Standard (Zymo Research Corporation, United States) for establishing the hybrid assembly pipeline using Illumina and ONT sequence. This mock community contains a mixture of eight bacterial strains (each contributing 12% of the total DNA) and two fungi (each accounts for 2% of total DNA). The nanopore data of Zymo-GridION-EVEN-BB-SN were downloaded from https://github.com/LomanLab/mockcommunity. Illumina raw sequencing reads were prepared using Rapid DNA Library Prep Kit and sequenced using NovaSeq (PE150).

Illumina raw sequences were quality-controlled using 'read_qc' module of MetaWRAP 1.2[29]. Nanopore raw sequences were base-called from fats5 files and quality-controlled using Guppy v.3.3.0 (ONT). For sequences from both platforms, human host reads were identified and removed by mapping against the human genome (hg19) with minimap2 v2.17-r941[30]. We compared five pipelines for metagenomic assemblies, including Canu 1.7[31], Flye 2.8.1-b1676[32], OPERA-MS[9], which only uses ONT reads; and hybrid assembly using Spades v3.13.0;[33] and only Illumina reads assembly of MetaSPAdes v3.13.0[33], to assemble from sequences of the mock community. The total length, contig numbers, largest contig length, N50, L50, average nucleotide identity (ANI) and run time were calculated to evaluate the assembly efficiency with Quast v.5.0.0[34] (Supplementary Fig. 1a). In addition, the abundance of eight bacterial species was assessed by Salmon 0.13.1[35] or using nanopore reads by minimap2[30]. The obtained contigs were then binned using MetaWRAP 1.2[29] to form MAGs. The MAGs with completeness >70% and contamination <10% were kept after refinement and reassemble. No MAG was obtained from the assembly of OPERA-MS. Although Flye predicted 14 contigs were circular, only two contigs were near the corresponding species in genome

length (92.68% and 86.65% respectively, Supplementary Data 1). At last, open-reading-frames (ORFs) were predicted using Prokka 1.13[36] and coding density was calculated (sum of length of ORFs divided by total length of MAG) for examining whether ONT reads introduced more errors into contigs and reduced ORFs numbers in a genome.

**Subject recruitment.** We recruited a total of 110 volunteers, including 100 healthy individuals without apparent diseases or infections, to form the cross-section cohort and each individual provided one fecal, one serum, and one urine sample on the same day; and 10 healthy individuals form the time-series cohort with 10 consecutive fecal and urine sampling. Additional information including age, body height and weight, blood pressure and fasting blood sugar level were recorded simultaneously with fecal samples (Supplementary Data 2 and 3). This study is approved by the ethic committee of Institute of Microbiology, Chinese Academy of Science with approval number APIMCAS2021003; all individuals were fully informed and have provided written consent, and compensated for traveling.

**Stool sample processing and sequencing.** Fecal samples were aliquoted into 2-ml cryovial tubes and placed at 4 °C immediately on collection, and transferred to storage at −80 °C within the same day. Fecal DNA was extracted with the Qiagen AllPrep PowerFecal DNA/RNA Kit (QIAGEN, Germany) using standard bead-beating mechanical lysis. The fecal DNA were separated to two parts for Illumina and ONT sequencing respectively. Illumina short-read libraries were prepared using NEXTflex™ Rapid DNA Library Prep Kit (Bioo Scientific, United States) and sequenced using Illumina NovaSeq (PE150). For ONT long-read libraries, fecal DNA was size-selected to remove DNA fragments < 5 kbp with BluePippin (Sage Science, United States), and libraries were prepared using Oxford Nanopore Technologies (ONT) Ligation library preparation kit (SQK-LSK109, EXP-NBD104, and EXP-NBD114) following manufacturer's instructions and sequenced with the ONT PromethION sequencer using FLO-PRO002 flow cells. ONT sequencing runs were scheduled for 48–60 h, and allowed to run until fewer than ten pores remained functional.

**Metabolome in fecal, serum, and urine samples.** The metabolites in all samples were identified and quantified by referring to the published studies as follows (with detailed information on metabolites provided in Supplementary Data 9):

(1) Sample processing and extraction of metabolites: All samples were collected and immediately refrigerated at −80 °C until preprocessed, and sample processing and metabolite extraction were referred to the study by dunn et al. and Gratton et al.[37,38]. The procedure was as follows:

(a) For fecal samples, 50 mg were transferred to an EP tube, and after adding 1000 μL extract solution (acetonitrile: methanol: water = 2: 2: 1, with 500 nM internal standard L-Leucine-5,5,5-d3 (Formula: C6H10D3NO2, MW:134.19, CAS: 87828-86-2)), samples were vortexed for 30 s and the samples were then homogenized at 35 Hz for 4 min and sonicated for 5 min in ice-water bath, a process repeated for 3 times Then the samples were incubated for 1 h at −40 °C and centrifuged at 10,800 $g$ for 15 min at 4 °C and the extract was transferred to a fresh glass vial for further analysis.

(b) for serum samples, 100 μL were extracted with 400 μL extract solution (acetonitrile: methanol = 1:1, with 500 nM internal standard L-Leucine-5,5,5-d3), and vortexed for 30 s, then sonicated for 10 min in ice-water bath and incubated for 1 h at −40 °C to precipitate proteins. Then samples were centrifuged at 10,800 $g$ for 15 min at 4 °C, and the extract was then transferred to a fresh glass vial for further analysis.

(c) For urine samples volume, urine was first normalized according to creatinine concentration, and 100 μL corrected urine were mixed with 400 μL of extract solution (acetonitrile: methanol = 1: 1, containing isotopically-labeled internal standard of 500 nM internal standard L-Leucine-5,5,5-d3), the mixture were vortexed for 30 s, sonicated for 10 min in ice-water bath, and incubated for 1 h at −40 °C to precipitate proteins. Then samples were centrifuged at 10,800 $g$ for 15 min at 4 °C. The quality control (QC) sample was prepared by mixing an equal aliquot of the supernatants from all of the samples.

(2) LC-MS/MS Analysis. LC-MS/MS analyses were performed using an UHPLC system (Thermo Fisher Scientific, SanJose, CA) with a UPLC BEH Amide column (2.1 mm × 100 mm, 1.7 μm, Waters, Manchester, UK) coupled to Q Exactive HFX mass spectrometer (Orbitrap MS, Thermo Fisher Scientific). Extracts were gradient-eluted with water (containing 25 mmol/L ammonium acetate and 25 mmol/L ammonia hydroxide, pH = 9.75) and acetonitrile. The mass spectrometry was used to acquire MS/MS spectra on information-dependent acquisition (IDA) mode in the control of the acquisition software (Xcalibur 4.0.27, Thermo Fisher Scientific). The ESI source conditions were set as following: sheath gas flow rate as 50 Arb, Aux gas flow rate as 10Arb, capillary temperature 320 °C, full MS resolution as 60,000, MS/MS resolution as 7500, collision energy as 10/30/60 in NCE mode, spray Voltage as 3.5 kV (positive), or −3.2 kV (negative), respectively.

(3) Data preprocessing and annotation. The acquired MS data pretreatments included peak selection and grouping, retention time correction, second peak grouping, and isotopes and adducts annotation, were performed as previously described[39]. LC-MS raw data files were converted into mzXML format and then

analyzed by the XCMS and CAMERA toolbox with R statistical language (v3.6.2). By using retention time and the m/z data pairs as the identifiers for each ion, we obtained ion intensities of each peak and generated a three-dimensional matrix containing arbitrarily assigned peak indices (retention time-m/z pairs), ion intensities (variables) and sample names (observations). Exacted molecular mass data (m/z) of peaks were searched through online HMDB database and KEGG database for metabolite identification. Exact molecular mass data (m/z) of peaks were searched through online HMDB database and KEGG database for metabolite identification. If a mass difference between observed and theoretical mass was < 10 ppm, the metabolite was annotated and the molecular formulas of the matched metabolites were further identified and validated by isotopic distribution measurements. Commercial reference standards were used to validate and confirm metabolites by comparing their MS/ MS spectra and retention time. The matrix was further reduced by removing peaks with missing values (ion intensity = 0) in more than 50% samples and those with isotope ions from each group to obtain consistent variables. Each retained peak was then normalized to the QC sample using Robust Loess Signal Correction (R-LSC) on the basis of the periodic analysis of the QC sample and the true samples to ensure the data of high quality within an analytical run, which is accepted as a quality assurance strategy in metabolic profiling. The relative s.d. (RSD) value of metabolites in the QC samples was set at a threshold of 30%, as a standard in the assessment of repeatability in metabolomics data sets.

**Hybrid assembly of human gut microbiome.** Quality control of raw sequences for Illumina and ONT reads from our human gut microbiome was carried out as described above and quality-filtered reads from each sample were assembled using MetaSpades v3.13.0[33] with -meta and -nanopore in parameters. Our choice of metaSpades for hybrid sequencing analysis was a result of balancing several parameters; compared to only using Illumina, hybrid assembly did indeed have a higher level of contamination, yet using Illumina alone could not achieve long contigs (as indicated by N50), which was especially important for the following up SVs analysis. The obtained contigs of each sample were then binned using Meta-WRAP 1.2[29] to form MAGs. The MAGs with completeness >70% and contamination < 10% were kept after refinement and reassemble; all MAGs were then combined and dereplicated using dRep v2.6.2[13] with the following options: '-pa 0.9 -sa 0.95 -nc 0.30 -cm larger'. The representative MAGs was picked based on a score, which was calculated for each genome using following formula:

$$Score = 1 * Completeness - 5 * Containmination + 0.5 * log(N50) \quad (1)$$

From there, rRNA presence in each MAG was determined with RNammer −1.2[40], the taxonomy information of dereplicated MAGs then were classified by gtdbtk 1.3.0[41]. For comparison of previous published Unified Human Gastrointestinal Genome (UHGG) collection, we downloaded their 4,644 nonredundant genomes. Then our hybrid MAGs and their 4,644 nonredundant genomes were pooled and were dereplicated using dRep v2.6.2[13] with the with the same options above. Then the numbers of shared genomes and novel genomes were calculated from this result. The MAGs that were assigned to high score and determined as representative MAGs were labeled as better genomes. The MAGs that were not found in UHGG dataset were labeled as novel genomes.

**SV analysis in assembled MAGs.** The SV events (insertion, deletion, and inversion) of each MAG (species) present in >10 samples were detected using representative MAGs (with highest completeness among all MAGs of the same species across samples) as reference using modified MUM&Co v2.4.2[42]. To further reduce the potential false positive discovery of SVs, the SV events with 10 bp of the start/end point of contigs in MAGs were not considered. The total SV events number between a query MAG and representative were then normalized by the genome size of MAG and compared in different bacterial phyla/families, as well as in cross-section and time-series cohort. Minimap2 v2.17-r941[30] was used to re-map ONT reads to reference MAG sequences and query MAGs, which we then visualized the mapping results using IGV 2.6.2[43] to establish the identity of SVs; a random set of 53 large SVs (> 500 bp) were selected for manual validation (Supplementary Fig. 2). Genes containing breaking points of predicted SVs were annotated by Prokka 1.13[36] and KEGG Orthology (KO) profiles were annotated by emapper.py 1.0.3[44,45]. KEGG enrichment analyses were performed using genes of predicted SVs as the foreground genes and all genes of all MAGs as the background. We only consider the SV in gene body region, because locating transcription start sites (TSSs) has been a bioinformatically challenging job[46], with best tools developed only for limited number of organisms such as E. coli (and limited types of transcription factors)[47,48], and even in this model organism the complexity of TSSs were overlooked until high-throughput screening techniques revealed many more non-classical TSSs[49]. In addition, our analytical focus on large SVs (>500 bp) means that even if we analyze TSS regions, gene bodies would almost surely be covered by the same SV.

**Correlation analysis of SVs and metabolome.** For each of the 189 MAGs (species) with >10 occurrences in our cross-sectional cohort, we first removed species with <50% occurrences in the cohort and then generated presence/absence matrix of SV-affected genes (grouped by KO) in the rest; then we pre-screened the

SVs to ensure that sufficient numbers of each group (SV and non-SV groups), and filtered either SVs or non-SVs with occurences below 20% of the total number. We also kept metabolites that are present in > 90% of fecal, serum, or urine samples. Spearman correlations were calculated between bacteria-metabolite pairs. Within significantly correlated bacteria-metabolite pairs (FDR < 0.1), we further tested spearman correlations considering SVs-affected genes (KOs) to discover subgroups (strains) of bacteria containing SVs (SV1 group) on such KO and with abolished correlations (*p* > 0.05) to the corresponding metabolite, while the subgroup without SVs (SV0 group) remain significantly correlated (*p* < 0.05) with the metabolite. The details of subgroups are available in the Supplementary Data 6. Among the bacterial species and metabolites, we examined also the potential confounding effects of anthropometric parameters, including age, gender, and BMI; for those significant association pairs involving metabolites correlated to one or more anthropometric parameters, we performed post-hoc analysis to determine the significance of bacteria-metabolite associations after controlling age/gender/BMI (Supplementary Data 8). Here we are cautious to remove the effects of anthropometric parameters in bacterial-metabolite associations due to the possibility that bacteria abundances might be driven by anthropometry and significant associations could be neglected after controlling for age/gender/BMI in metabolites and/or bacterial abundances.

**Prophage identification**. We used ProphageHunter (https://pro-hunter.genomics.cn/index.php/Home/hunter/hunter.html)[21], a novel integrative tool that employs both sequence similarity-based searches and prophage genetic features-based machine learning classification, to identify potentially active prophages in the contigs. Only those categories defined as active phages in the ProphageHunter results were selected as candidate prophages, and prophages found duplicated on the same contigs were removed. All remaining prophage candidates were pooled and further deduplicated using CD-hit v4.7[50] with 95% identity, with 9,805 non-redundant prophages in total as result. The prophages were further screened against at least one of the encoded major capsid proteins or terminase large subunit proteins, resulting in 2,247 prophages. Additionally, the active prophages were determined using by PropagAtE (Prophage Activity Estimator)[51], which uses genomic coordinates of integrated prophage sequences and short sequencing reads to estimate whether a given prophage is in the lysogenic (dormant) or lysogenic (active) infection phase.

**Phylogenetic and host analyses of prophages**. The phylogenetic trees of prophages were constructed based on the concatenated terminase large subunit (TLS) and major capsid protein (MCP), whose amino acid sequences of the coding protein annotated as terminase large subunits and major capsid proteins were extracted separately and simultaneously aligned using MAFFT v7.450[52] (-localpair -maxiterate 1,000), and the sequences that were poorly aligned were pruned. The phylogenetic tree was constructed using the IQTREE2[53] automatic selection model and 1000 bootstrap replicates, and visualized using iTOL v5[54]. In addition to prophages, the annotation of bacterial host from the flanking region was performed use CAT software[55].

**CRISPR array detection**. We used CRSPRDetect v2.4[56] to predict direct repeat sequences and spacer sequences with a cutoff value of 3 for the CRISPR likelihood score. To summarize, we identified the CRISPR arrays in our MAGs by using CRISPRDetect and the officially recommended parameter "-array_quality_score_cutoff = 3" (https://github.com/ambarishbiswas/CRISPRDetect_2.2). To further remove false positives, CRISPRDetect searched for CRISPR arrays with greater than 2 repeats, and putative CRISPRs with repeat lengths less than 20 were rejected. Using this criterion, we limited the false positive rate to ca. 0.79%, as calculated from the repeats shorter than the shortest repeats (23 nt) validated experimentally. CRISPRDetect Parser (https://github.com/hwalinga/crisprdetect-parser) was used to parse the output of CRISPRDetect to extract spacer sequences, resulting in 7446 repeat sequences and 150,058 spacer sequences. Additionally, we used blast v2.6.0[57] with a threshold of identity ≥80% and *e*-value ≥ 1e-5 to compare with CRISPROpenDB[24] and other spacers from western population[25]. To investigate the correlation between prophages and spacers, we aligned spacers with prophage sequences using blast v2.6.0[57] and allowed for mismatches of 2–3 bases.

**Statistics**. For testing the relationship between numbers of SV events and reference genome frequency, query genome size, query genome contamination and query genome completeness, we used linear regression model. For comparision of average number of SVs per 1 Mb genome between MAGs from different individuals in the cross-sectional cohort and from different samples within the same individual in the time-series cohort, we used two-sided Wilcoxon test. KEEG functional enrichment analysis was conducted based on the result of Fisher test of each SV-affected gene. For correlation analysis of SVs and metabolome, spearman correlations were calculated using corAndPvalue function from WGCNA v1.69[58] package and p-values were adjusted according to Benjamini-Hochberg method (FDR threshold 0.1) using mt.rawp2adjp function from multtest v2.38.0[59] package. For examing the potential confounding effects from anthropometric parameters, including age, gender, and BMI, post-hoc analysis was used to determine the significance of bacteria-metabolite associations after controlling age/gender/BMI. All above statistics were carried out in R v3.5.2[60]

**Reporting summary**. Further information on research design is available in the Nature Research Reporting Summary linked to this article.

## Data availability
ONT and Illumina sequencing data generated from this study is deposited in NCBI SRA database with Project ID: PRJNA820119. Source data are provided with this paper.

## Code availability
The scripts used for the analysis reported in this study are publicly available at https://github.com/chen318liang/Gut-Metagenome-Pipeline-Based-on-Nanopore-Sequencing.git.

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

## Acknowledgements

This study was funded by National Key Research and Development Program of China (2021YFA1301000 to J.W.), the National Natural Science Foundation of China (91857101 to J.W.), the Strategic Priority Research Program of the Chinese Academy of Sciences (XDB29020000 to J.W.), the Beijing Natural Science Foundation (JQ20031 to MS) and the State Key Laboratory of Membrane Biology (M.S.).

## Author contributions

Conceptualization: J.W. and M.S. Methodology: L.C., N.Z., J.C., X.L., and J.X. Investigation: L.C., N.Z., J.X., Y.M., Y.Y., X.Z., W.Z., X.G., L.Z., X.Y., Z.L., Y.F., D.W., Y.W., and F.L. Visualization: J.W., L.C., J.C., and X.L. Funding acquisition: J.W. and M.S. Project administration: L.C., N.Z., J.C., X.L., and J.X. Writing – original draft: J.W., L.C., J.C., and X.L. Writing – review and editing: J.W., M.S., N.Z., and L.C.

## Competing interests

The authors declare no competing interests.
