## [Peer Review File · Nature Communications]

REVIEWER COMMENTS

Reviewer #1 (Remarks to the Author):

As increasing studies support the important roles of gut microbiome in human health and diseases, these studies have also uncovered greater complexity of microbiome than previously understood. The widely used short read sequencing has limitations in resolving the complexity of metagenomes. The advent of long read sequencing technologies provides a new opportunity to address the limitations of short read sequencing. This manuscript attempted to use nanopore long read sequencing to study structural variations and epigenetic variations in healthy human gut microbiome. While studies like this one should definitely be encouraged, this manuscript has some major issues. In brief, the analyses of structural variations have some aspects that need to be improved and better described, which can probably be addressed in a revision. However, the DNA methylation analysis was not performed in the right way, for the reasons described below in detail, in hope of helping the authors in a constructive way. To do the methylation analysis in the right way, the authors would need to generate a large number of new sequencing data. Alternatively, the authors may consider to only focus on structural variations in this manuscript, which would still be a great contribution to the field after proper revision.

1. For SV analysis in cross-sectional cohort, the authors used the representative MAGs of the 189 high prevalent species (present in > 10 individuals) as references. Long read sequencing metagenomic assembly may still generate misassemblies and chimeric contigs, which can affect reference quality and subsequently SV analysis. So, it is important to systematically examine the assembly quality of those MAGs (e.g. contamination, completeness, etc). Although the method section mentioned checkM, detailed QC analyses were not provided either in maintext or supplementary material.

2. The authors mentioned that the time-resolved samples supported that the genome structure of the same species are largely stable, and that that the strain differentiation/replacement observed over three years could be results of gradual SV accumulations. However, based on previous longitudinal studies, the 10-day window is likely too short to represent actual gut microbiome stability. This can affect data interpretation in the downstream analyses.

3. Fig 1E, 1F, it is unclear why the distributions of insertion and deletion have two peaks, while the distribution of inversions have one peak only. Important to examine the data to confirm if there is any sequencing or analytical bias.

4. To examine the effect of SV on gene functions, did the authors only consider SVs in the gene body region? What about SVs that affect gene promoters?

5. Lines 222-226, it's interesting that "Agathobacter rectalis was significantly associated with fecal fructose-1-phosphate (F1P) when ignoring SVs ($R_{\text{spearman}} = 0.28$, $p = 0.0053$), among other bacteria. Further analysis accounting for the status of 33 SV-affected genes showed that the subgroup (strain) of bacteria containing SVs no longer had significant correlations with F1P (eg. K01193 in *Agathobacter rectalis*, $R_{\text{spearman}} = 0.18$, $p = 0.2$, Fig. 3D, 3E, 3F)." It would be helpful if the authors can indicate the sample size for each subgroup compared in the statistical tests.

6. This is likely the largest microbiome data generated to date using nanopore sequencing. To ensure broad impact, the authors would want to make sure to deposit the data with easy access, and may consider depositing the data in SRA in addition to NMDC.

7. The methylome analysis was unfortunately not performed correctly. The authors have a fundamental misunderstanding that 6mA events in bacteria are on GATC sites. This is not the case. Actually, the vast majority of bacterial species do not have 6mA on GATC sites, but on other diverse sequence motifs. The authors may find the following two reviews helpful

<https://www.nature.com/articles/s41579-019-0286-2>

<https://www.nature.com/articles/s41576-018-0081-3>

Due to this fundamental misunderstanding, the entire DNA methylation analyses and interpretation are not correct unfortunately. For example, Fig. 5B shows methylated GATC sites across different bacteria families, but most of them do not encode DNA methyltransferases that target GATC sites. This fundamental error in methylation analysis was also due to the tool used: Tombo, which was not designed for **de novo** DNA methylation analysis in bacteria, and it can generate substantial false positive calls. Currently, the only tool for reliable de novo methylation discovery from bacteria is nanodisco, <https://www.nature.com/articles/s41592-021-01109-3> which requires both matched native and whole genome amplification (WGA) data for each microbiome sample. If the authors hope to perform methylation in a reliable way, they would need to generate a large number of new WGA sequencing data. Alternatively, the authors may consider to only focus on structural variations in this manuscript, which (after revision) would still be a great contribution to the microbiome field.

Minor issues

1 Line 116, Supplementary Fig. 1B, 1C do not seem to support the statement here. Please double check it.

2 Line 118, according to the results, 623 out of 692 dereplicated MAGs corresponded to HUGG bins, the remaining 67 are novel bins. How about the other 2 MAGs?

3 Line 127, please check the sentence, "Expanding the scope of detected structural variations in gut microbiome".

4 Fig. 1A, the supporting membrane has negative charge in the upside and positive charge in the downside.

Reviewer: Gang Fang

Reviewer #2 (Remarks to the Author):

In this paper Chen et. present a relevant study which characterizes the structural variations and methylome profiles of more than a hundred gut microbiomes, providing also a longitudinal overview of these profiles in ten individuals. For this, authors implement a novel pipeline which relies on a combined assembly of short (Illumina Technology) and long sequencing reads (Oxford Nanopore Technology). They identified a high intra-individual stability of structural variations, but large variations across different subjects. The authors also determined the bacterial methylation profiles (meta-methylome), related it to dietary methionine intake, and validated these findings in a mouse model.

The main limitation of this study is the rather low sample size, which makes the analysis in relation to metabolomics and diseases underpowered. Similarly, the methylation study is limited by the assessment of only one methylation motif, and the low number of species in which the correlation with fecal metabolites and in vivo validation are assessed, which should be considered and discussed when interpreting the results.

However, this study includes one of the first approaches to incorporate ONT methods to the study of the gut microbiome. Authors present a novel methodology based on well-designed comparison of multiple assembly strategies. This hybrid method constitutes an improvement in terms of the scope of SVs detection in comparison to short-read sequencing based methods, allowing the detection of different types of SVs and their validation at the read level. Moreover, this approach represents a considerable improvement in the detection of prophages and CRISPR spacers, highlighting it as a suitable method for future studies aiming to expand the diversity of known CRISPR spacers and phage-bacterial host pairs. Finally, this study provides new insights into the highly unexplored stability and dynamics of microbial methylation profiles, representing a comprehensive pilot study on the field. The descriptive part of this analysis has already sufficient impact, and for that the sample size is sufficient.

Additional comments /improvements:

Main text:

1. In the section discussing the improvement of the quality of human gut metagenome assembly using the hybrid assembly approach, authors comment that the quality of 208 MAGs assembled with the hybrid pipeline is higher than the MAGs from HUGG database. However, it is not explained which metrics do they use for such comparison. Likewise, they do not provide information of these quality results for MAGs found by short-read approach. This should be discussed and included in the text.

2. Supplementary Figure 1D shows high quality assembly results for hybrid assembly using metaSPAdes, with high ANI and completeness and very low contamination. However, results from 1B show very similar results in the binning step in terms of completeness between this procedure and NGS using metaSPAdes, with the former showing a considerably higher contamination rate. However, this is not discussed in the text.

3. This sentence is unclear “We further examined whether ONT reads introduced more errors into contigs and reduced open-reading-frames (ORFs) numbers in a genome (i.e., coding density), and found that hybrid assemblies had no detectable reduction in coding density compared to that by Illumina-only assemblies, while using only ONT reads led to significant decrease (Supplementary Figure 1C).” – decrease of what?

4. In the section discussing the functionality of genes disrupted by SVs, the enrichment analysis is focused only on reference MAGs. Considering the strong inter-individual variation, the disruption pattern of genes could be different between individuals. Will the functionality disruption observed here be conserved among individuals? This should be discussed.

5. Deletions and insertions seems to have a clear bi-modal distribution (Figures 1E-1F), do the authors have suggestion on the nature of this distribution?

6. I disagree with the authors on the formulation that “SVs complicate bacterial-metabolite correlations” – if correlation with species is different in relation to presence/absence of SVs, this is not a “complication” but defines that presence of certain bacterial genes are relevant for the phenotype. Of course, it is clear that in the rather small number of samples (100 individuals investigated here) and the very large number of metabolites (which actually should be more clearly indicated in the paper), this can be a “complication” since it adds additional variables to already underpowered study. Similar statement is also in discussion: “Via metabolites such as neotrehalose and F1P, SVs complicated the correlations of bacterial abundances to host health phenotypes such as blood glucose.”

7. It is unclear which methods were used for multiple testing correction in the metabolomics study – the authors mention p-values, and did not describe correction method. Nominal p-value of 0.05 should not be considered a significant finding, please clarify.

8. “Among the metabolites and SV-affected genes, we found four metabolites affected by SVs and a total of 11 genes affected by SVs were mapped to four KEGG pathways, in which the SV-affected genes and metabolites were both involved, strongly suggesting the roles of SVs in shaping bacterial-

metabolite correlations by affecting the function of relevant genes.” Is it more than expected by chance, given the number of tested metabolites and SVs?

9. The authors identified >150,000 CRISPR spaces, most of which were not present in the existing databases. The more detailed description of this will be of interest, i.e. – what was the size distribution of spacers? How many of them do you expect to be false-positive findings?

10. The authors did not say if the analysis was corrected for age/gender/BMI and/or any other covariates.

11. Please, provide link to the pipeline (i.e. github) that was built to analyze the ONT data and to combine it with Illumina data, and other scripts used in this study.

12. Supplementary Figure 7 is unclear and does not illustrate the message from the text. Which names in the figure indicate bacterial genes from SVs? Which are reflecting human genes? What does it all mean? “The little circles rectangle in the diagram indicated compounds and genes involved in pathway, respectively. and the highlighted ones are our mapping input metabolite and genes.” What is “Little circles rectangle” and what is “the highlighted ones”. What is the Word “respectively” refers to? When pointing to the names, text in Chinese appears. Overall, the corresponding text in the results section is also not very clear.

Figures and minor points:

13. The references to figures start with Supplementary Figure 1D, while 1A is not referenced, and 1B, C are referenced afterwards.

14. Supplementary Figure 1 has a lot of abbreviations that are only explained later in the methods. It would be valuable to extend the legend for Supplementary figure 1 to explain the terms, i.e. which assemblies are only using the ONT data, which is Illumina and which one is hybrid. It is unclear why OPERA is not included to comparison in Supplementary Figure 1C.

15. Supplementary Figure 6 should be modified, as it is not possible to visualize regression line corresponding to the correlation without considering SVs.

16. The quality of some figures should be improved: Figures 4, 5; Supplementary Figures 4, 5, 6, 8, 10, 12.

17. Figure 1D: substitute “inseration” by “insertion”.

18. Supplementary Figure 4: Correct the name for Agathobacter rectalis.

19. Line 127: Sub-title “Expanding the scope of detecSupplementary Table tructural variations in gut microbiome” should be edited.

20. Line 246: Modify the sentence for clarity: “It has been reported that F1P can competitively inhibit the liver phosphorylase that metabolizes glycogen to glucose and thus potentially contributing to lowering blood glucose for trehalose it is not yet clear the relevance to blood glucose”

21. Line 261: The word “that” should be deleted: Furthermore, relying on long ONT reads we confirmed that the direct linkage between prophage elements and flanking host bacterial genomes”.

22. Line 308: Firmicutes name should be corrected: “... followed by Actinobacteriota and Firmicutes_A”.

23. Line 339: Modify the sentence for clarity and make it shorter: “Given that the essential amino acid methionine is necessary for synthesis of S-Adenosylmethionine (SAM) in the methionine cycle, the methyl donor required for Dam and other methyltransferases to methylate genomic DNA in bacteria, to verify the potential role of dietary methionine in modulating the methylation levels of gut microbiome, we performed feeding experiment in specific-pathogen-free mice with three types of diet”.

Reviewer #3 (Remarks to the Author):

The authors have used oxford nanopore sequencing with long reads to identify and increase coverage and binning of metagenomics and utilization of methylation patterns in large-scale human studies. I have the following comments.

1. Experimental setup of metabolomics analysis: The metabolomics section needs more rigor regarding annotation and quantification of metabolites. The authors cite references by Liu et al. (Ref 45), Dunn et al. (Ref 46) and Hou et al. (Ref 47), but these are largely inconsistent with each other. Liu et al. refers to Dunn et al. (also referenced in this paper as Ref 46), and a second paper by Hou et al. J Chromatogr A, 1429 (2016), pp. 207-217 that is also cited here. Hou et al. however only quantifies 11 metabolites using a Triple-Quadrupole targeted method, a fundamentally different method. Dunn et al. is a general standard protocol from more than 10 years ago that uses generic descriptions of GC-MS, and high resolution MS using either orbitrap or QTOF platforms. None of these papers describes metabolite extraction procedures for feces. The authors really need to describe their platform and workflow to prepare the samples, and to identify and quantify the data in sufficient detail, instead of referring to various previously published studies.

2. Chromatography: In this regard, it is concerning to this reviewer that the authors do not give details on the chromatography used. This is very important to be able to reproduce the analysis. It is surprising to this reviewer that some separation of very polar compounds (i.e. alanine, glycine, serine) was achieved using the HTSS-T3 column that is C18 based. I would have expected a more polar column (i.e. HILIC or BEH amide) to convincingly separate polar compounds. Retention times, exact masses and means to identify metabolites need to be accurately described.

3. Confidence of metabolite identification: The authors need to describe their procedure for identifying metabolites more accurately, including the strategies for metabolite identification.

4. Statistical analysis. The integration of metabolite signals (what signals were used?) with the methylation data need to be more accurately described.

5. Functional validation. The authors use a methionine feeding study that shows that methionine changes methylation (Figure 5E). It is, however, concerning to this reviewer that the control is very similar to the high methionine containing diet. I feel this experiment would benefit from some extra controls that show that really methylation is increased by methionine, and not just the composition of the microbiome. I was not able to conclude that from the figure and the Supplemental Figure 10-12 provided.

Additional comments:

The authors should describe in how far their procedure here is an advance because many papers have already described the use of ONT to identify and sequence the microbiome. In addition, methylation has been shown to be monitored before other groups. For instance, Tourancheau et al (Nature Methods 2021), a paper cited, succeeded in assigning methylation type 4mC, 5mC or 6mA using ONT sequencing as well. Here, only 6mA (Figure 3) is considered for further analyses.

In Figure 3B, C it is unclear what the color codes refer to. In Figure 3B, the color code is too small to discriminate. While the presentation as a rose diagram in Figure 3C is fancy, I think it is not very useful and a more simple presentation (i.e. as a bar chart) will be more effective.

Figure 3D-I Correlations may be spurious and p-values should be corrected for multiple testing.

Table S2: Fasting blood sugar level needs a unit.

Response to Reviewers

Reviewer #1 (Remarks to the Author):

As increasing studies support the important roles of gut microbiome in human health and diseases, these studies have also uncovered greater complexity of microbiome than previously understood. The widely used short read sequencing has limitations in resolving the complexity of metagenomes. The advent of long read sequencing technologies provides a new opportunity to address the limitations of short read sequencing. This manuscript attempted to use nanopore long read sequencing to study structural variations and epigenetic variations in healthy human gut microbiome. While studies like this one should definitely be encouraged, this manuscript has some major issues. In brief, the analyses of structural variations have some aspects that need to be improved and better described, which can probably be addressed in a revision. However, the DNA methylation analysis was not performed in the right way, for the reasons described below in detail, in hope of helping the authors in a constructive way. To do the methylation analysis in the right way, the authors would need to generate a large number of new sequencing data. Alternatively, the authors may consider to only focus on structural variations in this manuscript, which would still be a great contribution to the field after proper revision.

Response: We thank Prof. Fang for this critical and insightful evaluation of our work. We appreciate the comments on the analytical results and interpretation of the SV section, and addressed the questions below; and also taking the advice on methylation into account, since it is not practical to re-sequence almost all the samples within short term (and with our budget) we have removed the section on methylation analysis, and leave them for future studies.

1. For SV analysis in cross-sectional cohort, the authors used the representative MAGs of the 189 high prevalent species (present in > 10 individuals) as references. Long read sequencing metagenomic assembly may still generate misassemblies and chimeric contigs, which can affect reference quality and subsequently SV analysis. So, it is important to systematically examine the assembly quality of those MAGs (e.g. contamination, completeness, etc). Although the method section mentioned checkM, detailed QC analyses were not provided either in maintext or supplementary material.

Response: Thanks for this suggestion. We have added the related information for the 189 high prevalent MAGs in supplementary material (Supplementary Fig. 2 and Supplementary Table 4). The median of completeness, contamination of the 189 MAGs are 98.73% and, 0%, indicating high completeness and low contamination, while the N50, L50 and largest contig were 171,537nt, 6 and 401,913nt respectively.

Supplementary Figure 2. The assembly quality of 189 MAGs (present in > 10 individuals) using for structure variation analysis

Supplementary Table 4. The classification and assembly quality of 189 MAGs (present in > 10 individuals) used for structure variation analysis. (Please see the supplementary tables).

2. The authors mentioned that the time-resolved samples supported that the genome structure of the same species are largely stable, and that that the strain differentiation/replacement observed over three years could be results of gradual SV accumulations. However, based on previous longitudinal studies, the 10-day window is likely too short to represent actual gut microbiome stability. This can affect data interpretation in the downstream analyses.

Response: We appreciate the reviewer’s comment. Organizing a short term, continuous sampling was more practical when we designed the study, as tracking individuals and collecting samples along longer intervals and time window would risk a higher rate of drop out; plus, among the three studies available on gut microbiome structural variations, two were cross-sectional ^{1, 2}; and one study ³ compared samples collected 3 years apart, where they indeed demonstrated that the SVs had significant changes within the same individual. We would still be unable to determine an appropriate time window to separate “short” vs “long” term for structural variations and it was not foreseeable when we designed the study. We acknowledge the limitations as you pointed out, and now down-tuned several interpretations to be more accurately reflecting the time-window of our study, as well as potential limitations. Please see line 310-315 of the manuscript.

References:

1. Zeevi, D. et al. Structural variation in the gut microbiome associates with host health. *Nature* 568, 43-48 (2019).

2. Wang, D. et al. Characterization of gut microbial structural variations as determinants of human bile acid metabolism. *Cell Host Microbe*. 29(12):1802-1814 e5 (2021).
3. Chen, L. et al. The long-term genetic stability and individual specificity of the human gut microbiome. *Cell* 184, 2302-2315 e2312 (2021).

3. Fig 1E, 1F, it is unclear why the distributions of insertion and deletion have two peaks, while the distribution of inversions have one peak only. Important to examine the data to confirm if there is any sequencing or analytical bias.

Response: Thanks for your suggestion. We have examined the bi-modal distribution pattern of the insertion and deletion length and confirmed a bi-modular distribution; the inversions are overall very low in number thus a rarer SV type, we do not have sufficient observations to determine the distribution pattern yet. Regarding the two peaks, we hypothesized that the two peaks of SVs were results of different biological processes in prokaryotic genome, especially with regard to transposon/prophage and other mobile elements' activities. Thus, we analyzed randomly selected SVs within two peaks (within 140-160bp and 1050-1150bp, respectively), and predicted the prophage and extrachromosomal mobile genetic elements (eMGEs) using blastn based on the mMGE database ¹. Results indicated significant differences between SVs within of the two peaks and mobile elements are significantly higher in short SVs: prohages in short vs long SVs: deletion ($p = 2.82e-06$), insertion ($p = 2.93e-05$); and eMGEs in short SVs vs long SVs: deletion ($p = 4.385e-07$), insertion ($p = 0.0005129$, all with Wilcox test). We thus infer the short SV are more likely results of phage integration and other mobile elements compared to longer ones. Yet, as not all SVs have detectable mobile elements, this offers only a partial and plausible explanation; we presume that the other SVs are results of replication error or recombination events but mechanistic validations are not available from limited studies focusing on SVs in bacteria (or overall). We have added tentative discussion in line 122-134 of the manuscript.

Reference:

1. Senying Lai, Longhao Jia, Balakrishnan Subramanian, Shaojun Pan, Jinglong Zhang, Yanqi Dong, Wei-Hua Chen, Xing-Ming Zhao, mMGE: a database for human metagenomic extrachromosomal mobile genetic elements, *Nucleic Acids Research*, 2021, 49:D1 D783–D791.

4. To examine the effect of SV on gene functions, did the authors only consider SVs in the gene body region? What about SVs that affect gene promoters?

Response: We thank the reviewer for this crucial questions, and indeed we considered promoters while designing the study, but were not able to do so. There are several reason for this: 1) Locating transcription start sites (TSSs) has been a bioinformatically challenging job ¹, with best tools developed only for limited number of organisms such as *E. coli* (and limited types of transcription factors)^{2,3}, and even in this model organism the complexity of TSSs were overlooked until high-throughput screening techniques revealed many more non-classical TSSs ⁴; for gut microbial species with limited understanding of transcription, we felt not confident to predict TSS regions and cross-check with our SVs. 2) Assuming text-book knowledge that TSS areas are usually located within 300bp of the gene, and relatively small inter-genic region length in bacteria (median size ca.300bp)⁵, our analytical focus on large SVs (> 500bp) means that even if we analyze TSS regions, gene bodies would almost surely be covered by the same SV. 3) Many genes in bacteria are formed in clusters and operons, with shared promoters and TSS, analysis of TSS would also mean to add the additional complexity of predicting far-reaching effects of one TSS to the whole operon. With these rationales, we chose to only focus on gene bodies, which were consistent with three other mentioned studies on SVs ⁶⁻⁸. We have now added additional explanation in the methods section, please see line 583-588.

References:

1. Gordon, J.J., Towsey, M.W., Hogan, J.M., Mathews, S.A. & Timms, P. Improved prediction of bacterial transcription start sites. *Bioinformatics* 22, 142-148 (2006).
2. Towsey, M.W., Gordon, J.J. & Hogan, J.M. The prediction of bacterial transcription start sites using SVMs. *Int J Neural Syst* 16, 363-370 (2006).
3. Lai, H.Y. et al. iProEP: A Computational Predictor for Predicting Promoter. *Mol Ther Nucleic Acids* 17, 337-346 (2019).
4. Thomason, M.K. et al. Global Transcriptional Start Site Mapping Using Differential RNA Sequencing Reveals Novel Antisense RNAs in *Escherichia coli*. *J Bacteriol* 197, 18-28 (2015).
5. Molina, N. & van Nimwegen, E. Universal patterns of purifying selection at noncoding positions in bacteria. *Genome Research* 18, 148-160 (2008).
6. Chen, L. et al. The long-term genetic stability and individual specificity of the human gut microbiome. *Cell* 184, 2302-2315 e2312 (2021).
7. Zeevi, D. et al. Structural variation in the gut microbiome associates with host health. *Nature* 568, 43-48 (2019).
8. Wang, D.M. et al. Characterization of gut microbial structural variations as determinants of human bile acid metabolism. *Cell Host Microbe* 29, 1802-+ (2021).

5. Lines 222-226, it's interesting that "Agathobacter rectalis was significantly associated with fecal fructose-1-phosphate (F1P) when ignoring SVs (Rspearman = 0.28, p = 0.0053), among other bacteria. Further analysis accounting for the status of 33 SV-affected genes showed that the subgroup (strain) of bacteria containing SVs no longer had significant correlations with

F1P (eg. K01193 in *Agathobacter rectalis*, Rspearman = 0.18, p = 0.2, Fig. 3D, 3E, 3F).” It would be helpful if the authors can indicate the sample size for each subgroup compared in the statistical tests.

Response: Many thanks for your comments. We have pre-screened the SVs to ensure that sufficient numbers of each group (SV and non-SV groups), and filtered either SVs or non-SVs with occurrences below 20% of the total number (see Methods, line 592-594). Currently, the sample sizes for the subgroups based on SV K01193|*A. rectalis* are $N_{SV0}=46$ and $N_{SV1}=54$, respectively; and the sample sizes for the subgroups based on SV K03655|*F. saccharivorans* are $N_{SV0}=48$ and $N_{SV1}=52$, respectively. We have now added the sample sizes of these two SV-based subgroups to the legend of Figure 3 (lines 398-421) In addition, we have also added information of subgroups involved in all significant association pairs to Supplementary Table 6.

6. This is likely the largest microbiome data generated to date using nanopore sequencing. To ensure broad impact, the authors would want to make sure to deposit the data with easy access, and may consider depositing the data in SRA in addition to NMDC.

Response: Thanks, we have deposited all sequences to NCBI SRA database with Project ID: PRJNA820119 (<https://www.ncbi.nlm.nih.gov/bioproject/PRJNA820119>), and revised this in line 451-452.

7. The methylome analysis was unfortunately not performed correctly. The authors have a fundamental misunderstanding that 6mA events in bacteria are on GATC sites. This is not the case. Actually, the vast majority of bacterial species do not have 6mA on GATC sites, but on other diverse sequence motifs. The authors may find the following two reviews helpful <https://www.nature.com/articles/s41579-019-0286-2> <https://www.nature.com/articles/s41576-018-0081-3>. Due to this fundamental misunderstanding, the entire DNA methylation analyses and interpretation are not correct unfortunately. For example, Fig. 5B shows methylated GATC sites across different bacteria families, but most of them do not encode DNA methyltransferases that target GATC sites. This fundamental error in methylation analysis was also due to the tool used: Tombo, which was not designed for **de novo** DNA methylation analysis in bacteria, and it can generate substantial false positive calls. Currently, the only tool for reliable de novo methylation discovery from bacteria is nanodisco, <https://www.nature.com/articles/s41592-021-01109-3> which requires both matched native and whole genome amplification (WGA) data for each microbiome sample. If the authors hope to perform methylation in a reliable way, they would need to generate a large number of new WGA sequencing data. Alternatively, the authors may consider to only focus on structural variations in this manuscript, which (after revision) would still be a great contribution to the microbiome field.

Response: We thank the reviewer for this important insight and suggestion. As mentioned at the beginning, we now removed this section from revision following your advice. We want to re-iterate that, we were indeed aware that Tombo's performance was only reliable for one motif (even evident in *E. coli* benchmark data) and not for motif-independent discovery, as explained in the methods section and results; we then only focused on this reliable motif and acknowledged that other methylations sites might have been missed. But, since this would mean an unknown proportion of methylations are unknown in gut microbiome, we feel indeed the results are currently immature for the microbiome field to follow up on, and removing it is a better choice for the paper – resequencing MDA-amplified samples is not financially feasible. We hope that as the methodology improves, we can re-utilize this part of data and generate better understanding of gut meta-methylome in the future.

Minor issues

8. Line 116, Supplementary Fig. 1B, 1C do not seem to support the statement here. Please double check it.

Response: Apologize for this error, we have removed the wrong reference to figure.

9. Line 118, according to the results, 623 out of 692 dereplicated MAGs corresponded to HUGG bins, the remaining 67 are novel bins. How about the other 2 MAGs?

Response: Thanks for pointing this out. There were two MAGs fewer after dereplication, as after pooling our MAGs and those from HUGG, and used dRep to remove redundant MAGs. Reason for this is that there were relatively close MAGs in our cohorts and HUGG that were previously not clustered together (as they used dRep v2.2.4) are now clustered with a higher version of dRep (we used v2.6.2). We added explanation now at line 95-98.

10. Line 127, please check the sentence, “Expanding the scope of detected structural variations in gut microbiome”.

Response: Thanks. We have revised it by “Expanding the scope of detected structural variations in gut microbiome” in line 107.

11. Fig. 1A, the supporting membrane has negative charge in the upside and positive charge in the downside

Response: Apologize for this oversight, we have now corrected this error.

Reviewer: Gang Fang

Reviewer #2 (Remarks to the Author):

In this paper Chen et. present a relevant study which characterizes the structural variations and methylome profiles of more than a hundred gut microbiomes, providing also a longitudinal overview of these profiles in ten individuals. For this, authors implement a novel pipeline which relies on a combined assembly of short (Illumina Technology) and long sequencing reads (Oxford Nanopore Technology). They identified a high intra-individual stability of structural variations, but large variations across different subjects. The authors also determined the bacterial methylation profiles (meta-methylome), related it to dietary methionine intake, and validated these findings in a mouse model.

The main limitation of this study is the rather low sample size, which makes the analysis in relation to metabolomics and diseases underpowered. Similarly, the methylation study is limited by the assessment of only one methylation motif, and the low number of species in which the correlation with fecal metabolites and in vivo validation are assessed, which should be considered and discussed when interpreting the results. However, this study includes one of the first approaches to incorporate ONT methods to the study of the gut microbiome. Authors present a novel methodology based on well-designed comparison of multiple assembly strategies. This hybrid method constitutes an improvement in terms of the scope SVs detection in comparison to short-read sequencing based methods, allowing the detection of different types of SVs and their validation at the read level. Moreover, this approach represents a considerable improvement in the detection of prophages and CRISPR spacers, highlighting it as a suitable method for future studies aiming to expand the diversity of known CRISPR spacers and phage-bacterial host pairs. Finally, this study provides new insights into the highly unexplored stability and dynamics of microbial methylation profiles, representing a comprehensive pilot study on the field. The descriptive part of this analysis has already sufficient impact, and for that the sample size is sufficient.

Response: We appreciate the reviewers' overall evaluation and critical assessments. We acknowledge that the study contains samples that are not high in terms of microbiome study standards used Illumina reads, but we want to stress that ONT as a relatively new and expensive platform, per sample cost is ca. 10 times higher than that of Illumina, and achieving 200 samples have already been a giant step forward—a point also recognized by the reviewer. Regarding metabolites, we focused on common metabolites (appearing in > 90% individuals were analyzed) and limited the number to ca 200-400; and for methylation analysis we also focused on most abundant species thus limited to a few. As explained to reviewer 1 above and following suggestion, we removed the methylation results for now and will work on re-analyzing it in the future when remaining technical issues are solved.

Additional comments /improvements:

Main text:

1. In the section discussing the improvement of the quality of human gut metagenome assembly using the hybrid assembly approach, authors comment that the quality of 208 MAGs assembled with the hybrid pipeline is higher than the MAGs from HUGG database.

However, it is not explained which metrics do they use for such comparison. Likewise, they do not provide information of these quality results for MAGs found by short-read approach. This should be discussed and included in the text.

Response: Thanks for your suggestion. We have revised the main text to clearly clarify the metrics used for comparison, for those 208 “better” MAGs we used three measures: higher completeness, lower contamination, and higher N50. Following your advice, we have also added quality parameters for our benchmark in Supplementary table 4, in addition to the figure below.

Figure: Improved completeness, lower contamination and higher N50 in 208 MAGs by our method, in comparison to those already in UHGG.

2. Supplementary Figure 1D shows high quality assembly results for hybrid assembly using metaSPAdes, with high ANI and completeness and very low contamination. However, results from 1B show very similar results in the binning step in terms of completeness between this procedure and NGS using metaSPAdes, with the former showing a considerably higher contamination rate. However, this is not discussed in the text.

Response: We thank the reviewer for this comment. Our choice of metaSpades for hybrid sequencing analysis was a result of balancing several parameters; compared to only using Illumina, hybrid assembly did indeed have a higher level of contamination, yet using Illumina alone could not achieve long contigs (as indicated by N50), which was especially important for the following up SVs analysis. We have now added the discussion in line 554-557.

3. This sentence is unclear “We further examined whether ONT reads introduced more errors into contigs and reduced open-reading-frames (ORFs) numbers in a genome (i.e., coding density), and found that hybrid assemblies had no detectable reduction in coding density compared to that by Illumina-only assemblies, while using only ONT reads led to significant decrease (Supplementary Figure 1C). “ – decrease of what?

Response: Thanks for pointing out, we meant coding density and have revised in the manuscript (line 78).

4. In the section discussing the functionality of genes disrupted by SVs, the enrichment analysis is focused only on reference MAGs. Considering the strong inter-individual variation, the disruption pattern of genes could be different between individuals. Will the functionality disruption observed here be conserved among individuals? This should be discussed.

Response: Thanks for your suggestion, here we want to emphasize that the profiling and enrichment is at **population-level (i.e. across the whole cohort)**. Using the most complete MAG as reference is, firstly more computationally feasible to infer SVs in other individuals and indeed we observed inter-individual variations (as shown in figures below); but to keep the analysis meaningful and practical, we did not perform per individual level SV enrichment as the large amount yet non-representative calculations; and the relatively small amount of SVs in each individual is difficult for SV enrichment analysis (per individual the average number of SVs is 16.7 (per Mb genome) per MAGs and not suitable to conclude on pathway conservativeness). We have discussed in the revision and emphasized that the conclusions are for the population-level SVs, and acknowledged the large inter-individual variations (line 176-180).

Figure: Ten MAGs from same species were randomly selected and compared with the reference to show inter-individual differences regarding the structure variations. In circos plot, the red and blue arc indicated insertion and deletion, respectively. The length of arc indicated the size of each structure variation events. We zoomed out several representative SV regions in the IGV plot below to show details.

5. Deletions and insertions seems to have a clear bi-modal distribution (Figures 1E-1F), do the authors have suggestion on the nature of this distribution?

Response: Thanks for your suggestion, as also mentioned by reviewer 1. We have examined the bi-modal distribution pattern of the insertion and deletion length and confirmed a bi-modal distribution, and we hypothesized that the two peaks of SVs were results of different biological processes in prokaryotic genome, especially with regard to transposon/prophage and other mobile elements' activities. Thus, we analyzed randomly selected SVs within two peaks (within 140-160bp and 1050-1150bp, respectively), and predicted the prophage and extrachromosomal mobile genetic elements (eMGEs) using blastn based on the mMGE database (Lai et al. 2021; Nucleic Acids Research). Results indicated significant differences between SVs within of the two peaks and mobile elements are significantly higher in short SVs: prophages in short vs long SVs: deletion ($p = 2.82e-06$), insertion ($p = 2.93e-05$); and eMGEs in short SVs vs long SVs: deletion ($p = 4.385e-07$), insertion ($p = 0.0005129$, all with Wilcox test). We thus infer the short SV are more likely results of phage integration and other mobile elements compared to longer ones. Yet, as not all SVs have detectable mobile elements, this offers only a partial and plausible explanation; we presume that the other SVs are results of replication error or recombination events but

mechanistic validations are not available from limited studies focusing on SVs in bacteria (or overall). We have added tentative discussion in line 122-134.

6. I disagree with the authors on the formulation that “SVs complicate bacterial-metabolite correlations” – if correlation with species is different in relation to presence/absence of SVs, this is not a “complication” but defines that presence of certain bacterial genes are relevant for the phenotype. Of course, it is clear that in the rather small number of samples (100 individuals investigated here) and the very large number of metabolites (which actually should be more clearly indicated in the paper), this can be a “complication” since it adds additional variables to already underpowered study. Similar statement is also in discussion: “Via metabolites such as neotrehalose and F1P, SVs complicated the correlations of bacterial abundances to host health phenotypes such as blood glucose.”

Response: Thank you for the comment, and now we think the complication might be better presented in the form of “confounding”, which also reflects better as reviewer suggested per relevance to the phenotype. Certainly, 100 individuals are not great in terms of statistical power, but as explained above it is mainly due to the cost of ONT sequencing, not metabolomes. We in this analysis, have filtered the metabolites appearing in >90% of the individuals, and limited the analysis to 458 fecal metabolites, 286 serum metabolites and 396 urine metabolites, range common to most of the metabolome studies¹; further, as we controlled tightly on the false positive rates (to 0.1 in terms of bacteria – metabolites association, answering also the question below), and provided further related information from KEGG mapping, we believe the analysis do reflect the functional differentiations in bacterial strains due to SVs (and with respect to metabolites as phenotype read-outs). Our referred studies with Illumina sequencing also performed similar analysis in larger cohorts and concluded similar findings²⁻⁴. And for the statement mentioned by the reviewer, we now revised to “It is thus likely that via modulating the associations between bacteria and metabolites such as neotrehalose and F1P, SVs confound the correlations of bacteria to metabolites and eventually to important host phenotypes (such as blood glucose), adding a layer of complexity in association between gut microbiome and host health. Yet, further functional experiments are warranted to establish the findings as our analysis are still limited to correlation inference.” (line 322-326), in order to better clarify our point, as well as to acknowledge the current limitation.

References:

1. Bar, N., Korem, T., Weissbrod, O. et al. A reference map of potential determinants for the human serum metabolome. *Nature* 588, 135–140 (2020).
2. Chen, L. et al. The long-term genetic stability and individual specificity of the human gut microbiome. *Cell* 184, 2302-2315 e2312 (2021).
3. Zeevi, D. et al. Structural variation in the gut microbiome associates with host health. *Nature* 568, 43-48 (2019).
4. Wang, D.M. et al. Characterization of gut microbial structural variations as determinants of human bile acid metabolism. *Cell Host Microbe* 29, 1802-+ (2021).

7. It is unclear which methods were used for multiple testing correction in the metabolomics study – the authors mention p-values, and did not describe correction method. Nominal p-value of 0.05 should not be considered a significant finding, please clarify.

Response: Thank you for this critical insight. The significance of bacteria-metabolite association pairs corrected for multiple testing the Benjamini-Hochberg method (FDR threshold 0.1) (see Methods, line 599). In addition, we have screened the metabolite list prior to association analysis, removing a large number of unidentified metabolites and those occur in less than 90% of individuals, resulting in a calculation involving 458 fecal metabolites, 286 serum metabolites and 396 urine metabolites. For the example of SVs in the correlation of bacterial abundance and blood glucose, now we emphasize the better example of *F. saccharivorans*|K03655, whereas the SV0 group has significant association with blood glucose ($p=8.6e-5$, $FDR=0.001$). We have now added the multiple testing information to the legend of Figure 3 (lines 399-424), Supplementary Figure 6 and Supplementary Figure 7 and modified in the manuscript line (235-241).

8. “Among the metabolites and SV-affected genes, we found four metabolites affected by SVs and a total of 11 genes affected by SVs were mapped to four KEGG pathways, in which the SV-affected genes and metabolites were both involved, strongly suggesting the roles of SVs in shaping bacterial-metabolite correlations by affecting the function of relevant genes.” Is it more than expected by chance, given the number of tested metabolites and SVs?

Response: Many thanks for your comments. Although we found a total of 753 such gene-fecal metabolite association pairs, only 31 of the 74 metabolites involved are annotated in KEGG, and 46 of the 56 SV-affected genes were included in KEGG pathways (550 in total). The combinations of 4 metabolites (out of 31) and 11 genes (out of 46) co-locating in 4 metabolite pathways is ${}^{31}C_4 * {}^{46}C_{11} * {}^{550}C_4$, out of total number of scenarios where the metabolites and SVs randomly located in one pathway ($550^{31} * 550^{46}$), resulting in a probability of this being by chance at $1.5e-187$. Plus, the functional annotations in KEGG pathways gave additional support for the functional connections between genes and metabolites.

9. The authors identified >150,000 CRISPR spaces, most of which were not present in the existing databases. The more detailed description of this will be of interest, i.e. – what was the size distribution of spacers? How many of them do you expect to be false-positive findings?

Response: We thank the reviewer for the suggestion. To summarize, we identified the CRISPR arrays in our MAGs by using CRISPRDetect and the officially recommended parameter “-array_quality_score_cutoff = 3” (https://github.com/ambarishbiswas/CRISPRDetect_2.2). To further remove false positives, CRISPRDetect searched for CRISPR arrays with greater than 2 repeats, and putative CRISPRs with repeat lengths less than 20 were rejected. By 100% similarity de-redundancy, we obtained a total of 150,058 spacers with an average of 1665 ± 560 (mean \pm SD) spacers per metagenomic sample with average length of these spacers was 34 ± 4.8 nt (shown now in figure S9). Since the shortest length of experimentally validated CRISPR repeats is approximately 23 nt, a portion of our predicted repeats are even shorter in length, thus the positive rate was round 0.79%. We have added the relevant information in the methods section (line 633-638).

Supplementary Figure 9. Length distribution of CRISPR spacers. The graph shows the number of spacers (Occurrence > 500) with lengths from 27nt to 43nt, with vertical coordinates in thousands.

10. The authors did not say if the analysis was corrected for age/gender/BMI and/or any other covariates.

Response: We sincerely thank the reviewer for this comment, before the analysis of bacteria-metabolite associations, we tested the associations of bacteria and metabolites to age/gender/BMI, finding that only a minor proportion of gut bacterial species (7 out of 443, < 2%) and fecal metabolites (22 out of 458, < 5%) were significantly associated with one or more anthropometric parameters (with FDR < 0.1); and after establishing associations

between bacterial species and fecal metabolites, a retrospective analysis indicated that none of anthropometric-associated bacteria were in the significant association pairs (a total of 745), and only 4 out of 74 fecal metabolites involved in such pairs were associated with anthropometric parameters, involved in 31 pairs (*ca.* 4%); we none-the-less tested the effect of controlling for age/gender-BMI on those metabolites and found that 26 pairs remained significant ($p < 0.05$, the other 5 pairs have a p -value of 0.06), and now provided as additional information in Supplementary Table 8.

In terms of urine and serum metabolites, a higher proportion (185 out of 396 of urinary metabolites, and 79 out of 286 of serum metabolites) were significantly associated with one or more anthropometric parameters; and these included 28 metabolites in 60 bacteria-urine metabolite associations (out of 132), one metabolite in bacteria-serum metabolite pair (out of 2). We now also provide age/gender/BMI correction results for those pairs. Since the majority of associations were found in fecal metabolites to bacteria, from which we chose representative cases in the main text, the influences of age/gender/BMI were not substantial to our conclusions. Plus, we are also cautious for cases where the bacteria-metabolite associations are true, yet while correcting for anthropometric parameters we encounter cases that bacteria abundances and metabolites were both associated (and potentially driven) by age/gender/BMI, using the residues for correlational analysis after controlling for such effect would actually lead to false negative findings. We have now also added more discussion in the methods section (line 603-611)

11. Please, provide link to the pipeline (i.e. github) that was built to analyze the ONT data and to combine it with Illumina data, and other scripts used in this study.

Response: We have submitted the analysis pipeline and related script to the github (<https://github.com/chen318liang/Gut-Metagenome-Pipeline-Based-on-Nanopore-Sequencing.git>).

12. Supplementary Figure 7 is unclear and does not illustrate the message from the text. Which names in the figure indicate bacterial genes from SVs? Which are reflecting human genes? What does it all mean? “The little circles rectangle in the diagram indicated compounds and genes involved in pathway, respectively. and the highlighted ones are our mapping input metabolite and genes.” What is “Little circles rectangle” and what is “the highlighted ones”. What is the Word “respectively” refers to? When pointing to the names, text in Chinese appears. Overall, the corresponding text in the results section is also not very clear.

Response: Thanks for your suggestions and apologize for this error. We have modified Supplementary Figure 7 (now Supplementary Figure 8) and the legend accordingly, including the indication of the bacterial genes affected by SV, and the associated metabolites.

Figures and minor points:

13. The references to figures start with Supplementary Figure 1D, while 1A is not referenced, and 1B, C are referenced afterwards.

Response: Apologize. We have referenced this figure on line 71.

14. Supplementary Figure 1 has a lot of abbreviations that are only explained later in the methods. It would be valuable to extend the legend for Supplementary figure 1 to explain the terms, i.e. which assemblies are only using the ONT data, which is Illumina and which one is hybrid. It is unclear why OPERA is not included to comparison in Supplementary Figure 1C.

Response: Thanks, the figure legend has been expanded to include all the abbreviations. We did not get any bins from the contigs assembled by OPERA thus it was not included in Supplementary Figure 1C, now also explained in figure legend.

15. Supplementary Figure 6 should be modified, as it is not possible to visualize regression line corresponding to the correlation without considering SVs.

Response: Apologize, Supplementary Figure 6 (now Supplementary Figure 7) has been improved to be better visualized.

16. The quality of some figures should be improved: Figures 4, 5; Supplementary Figures 4, 5, 6, 8, 10, 12.

Response: Thank you for your suggestion. We have revised these figures to improve the resolution and PDF versions are available as supplemental zip.

17. Figure 1D: substitute “inseration” by “insertion”.

Response: Thanks. We have revised it.

18. Supplementary Figure 4: Correct the name for Agathobacter rectalis.

Response: Thanks. We have revised it.

19. Line 127: Sub-title “Expanding the scope of detecSupplementary Table tructural variations in gut microbiome” should be edited.

Response: Thanks. We have corrected it in line 107.

20. Line 246: Modify the sentence for clarity: “It has been reported that F1P can competitively inhibit the liver phosphorylase that metabolizes glycogen to glucose and thus potentially contributing to lowering blood glucose for trehalose it is not yet clear the relevance to blood glucose”

Response: Thank you for pointing this out. We have modified to “It has been reported that F1P can competitively inhibit the liver phosphorylase, which metabolizes glycogen to glucose, and thus potentially contributing to lowering blood glucose; for trehalose however, it is not yet clear the relevance to blood glucose.” Line 241-244.

21. Line 261: The word “that“ should be deleted: Furthermore, relying on long ONT reads we confirmed that the direct linkage between prophage elements and flanking host bacterial genomes”.

Response: Apologize. We have deleted it line 255.

22. Line 308: Firmicutes name should be corrected: “... followed by Actinobacteriota and Firmicutes_A”.

Response: Thanks a lot. However, the methylation analysis is now removed and this sentence is no longer in the revision.

23. Line 339: Modify the sentence for clarity and make it shorter: “Given that the essential amino acid methionine is necessary for synthesis of S-Adenosylmethionine (SAM) in the methionine cycle, the methyl donor required for Dam and other methyltransferases to methylate genomic DNA in bacteria, to verify the potential role of dietary methionine in modulating the methylation levels of gut microbiome, we performed feeding experiment in specific-pathogen-free mice with three types of diet”.

Response: Thank you for the suggestion, the methylation analysis is now removed and this sentence is no longer in the revision.

Reviewer #3 (Remarks to the Author):

The authors have used Oxford Nanopore sequencing with long reads to identify and increase coverage and binning of metagenomics and utilization of methylation patterns in large-scale human studies. I have the following comments.

1. Experimental setup of metabolomics analysis: The metabolomics section needs more rigor regarding annotation and quantification of metabolites. The authors cite references by Liu et al. (Ref 45), Dunn et al. (Ref 46) and Hou et al. (Ref 47), but these are largely inconsistent with each other. Liu et al. refers to Dunn et al. (also referenced in this paper as Ref 46), and a second paper by Hou et al. *J Chromatogr A*, 1429 (2016), pp. 207-217 that is also cited here. Hou et al. however only quantifies 11 metabolites using a Triple-Quadrupole targeted method, a fundamentally different method. Dunn et al. is a general standard protocol from more than 10 years ago that uses generic descriptions of GC-MS, and high resolution MS using either Orbitrap or QTOF platforms. None of these papers describes metabolite extraction procedures for feces. The authors really need to describe their platform and workflow to prepare the samples, and to identify and quantify the data in sufficient detail, instead of referring to various previously published studies.

Response: We thank the reviewer's insights and would like to confer our sincere apologies, as our major analysis was on the sequencing analysis, we failed to update the reference and details in our methods section regarding the generation of metabolome data, thus it contained outdated reference and technical inaccuracies. In the revision, we have updated the references for metabolome and details regarding different types of metabolome analysis. We hope the current revision resolves many issues from the first version, and again, we are deeply sorry for not presenting it accurately.

1) Sample processing and extraction of metabolites: All samples were maintained at -80°C until preprocessed according to previously described^{1,2} :

a) For fecal samples, 50 mg were transferred to an EP tube, and after adding 1,000 µL extract solution (acetonitrile: methanol: water = 2: 2: 1, with 500nM internal standard L-Leucine-5,5,5-d₃ (Formula: C₆H₁₀D₃NO₂, MW:134.19, CAS : 87828-86-2)), samples were vortexed for 30 s and the samples were then homogenized at 35 Hz for 4 min and sonicated for 5 min in ice-water bath, a process repeated for 3 times. Then the samples were incubated for 1 h at -40 °C and centrifuged at 12,000 rpm for 15 min at 4 °C and the extract was transferred to a fresh glass vial for further analysis.

b) for serum samples, 100 µL were extracted with 400 µL extract solution (acetonitrile: methanol =1:1, with 500nM internal standard L-Leucine-5,5,5-d₃), and vortexed for 30 s, then sonicated for 10 min in ice-water bath and incubated for 1 h at -40 °C to precipitate proteins. Then samples were centrifuged at 12,000 rpm for 15 min at 4 °C, and the extract was then transferred to a fresh glass vial for further analysis.

c) For urine samples volume, urine was first normalized according to creatinine concentration, and 100µL corrected urine were mixed with 400 µL of extract solution

(acetonitrile: methanol = 1: 1, containing isotopically-labelled internal standard (500nM L-Leucine-5,5,5-d3), the mixture were vortexed for 30 s, sonicated for 10 min in ice-water bath, and incubated for 1 h at -40 °C to precipitate proteins. Then the samples were centrifuged at 12,000 rpm for 15 min at 4 °C.

The quality control (QC) sample was prepared by mixing an equal aliquot of the supernatants from all of the samples.

2) LC-MS/MS Analysis. LC-MS/MS analyses were performed using an UHPLC system (Thermo Fisher Scientific, SanJose, CA) with a UPLC BEH Amide column (2.1 mm × 100 mm, 1.7 µm, Waters, Manchester, UK) coupled to Q Exactive HFX mass spectrometer (Orbitrap MS, Thermo Fisher Scientific). Extracts were gradient-eluted with water (containing 25 mmol/L ammonium acetate and 25 mmol/L ammonia hydroxide, pH = 9.75) and acetonitrile. The mass spectrometry was used to acquire MS/MS spectra on information-dependent acquisition (IDA) mode in the control of the acquisition software (Xcalibur 4.0.27, Thermo Fisher Scientific). The ESI source conditions were set as following: sheath gas flow rate as 50 Arb, Aux gas flow rate as 10Arb, capillary temperature 320 °C, full MS resolution as 60000, MS/MS resolution as 7500, collision energy as 10/30/60 in NCE mode, spray Voltage as 3.5 kV (positive) or -3.2 kV (negative), respectively.

3) Data preprocessing and annotation. The acquired MS data pretreatments included peak selection and grouping, retention time correction, second peak grouping, and isotopes and adducts annotation, were performed as previously described³. LC-MS raw data files were converted into mzXML format and then analyzed by the XCMS and CAMERA toolbox with R statistical language (v3.6.2). By using retention time and the m/z data pairs as the identifiers for each ion, we obtained ion intensities of each peak and generated a three dimensional matrix containing arbitrarily assigned peak indices (retention time-m/z pairs), ion intensities (variables) and sample names (observations). Exacted molecular mass data (m/z) of peaks were searched through online HMDB database and KEGG database for metabolite identification. The matrix was further reduced by removing peaks with missing values (ion intensity = 0) in more than 50% samples and those with isotope ions from each group to obtain consistent variables. Each retained peak was then normalized to the QC sample using Robust Loess Signal Correction (R-LSC) on the basis of the periodic analysis of the QC sample and the true samples to ensure the data of high quality within an analytical run, which is accepted as a quality assurance strategy in metabolic profiling. The relative s.d. (RSD) value of metabolites in the QC samples was set at a threshold of 30%, as a standard in the assessment of repeatability in metabolomics data sets.

We have now added in the methods section and also updated the references (line 503-550)

References:

1. Jasmine Gratton, et al. Optimized Sample Handling Strategy for Metabolic Profiling of Human Feces. *Anal Chem.* 2016; 88(9):4661-8.

2. Dunn WB, Broadhurst D, Begley P, et al. Procedures for large-scale metabolic profiling of serum and plasma using gas chromatography and liquid chromatography coupled to mass spectrometry. *Nat Protoc* 2011; 6(7): 1060-83.
3. Liu, R. et al. Gut microbiome and serum metabolome alterations in obesity and after weight-loss intervention. *Nat Med* 23, 859-868 (2017).

2. Chromatography: In this regard, it is concerning to this reviewer that the authors do not give details on the chromatography used. This is very important to be able to reproduce the analysis. It is surprising to this reviewer that some separation of very polar compounds (i.e. alanine, glycine, serine) was achieved using the HTSS-T3 column that is C18 based. I would have expected a more polar column (i.e. HILIC or BEH amide) to convincingly separate polar compounds. Retention times, exact masses and means to identify metabolites need to be accurately described.

Response: We thank the reviewer for raising this important question, and as indicated above, we are sorry for this inaccuracies described in the first version of manuscript. We indeed used BEH Amide column as now stated in the methods section. MS/MS spectra data was obtained on information-dependent acquisition (IDA) mode in the control of the acquisition software (Xcalibur 4.0.27, Thermo Fisher Scientific). Retention time and the m/z data pairs were used as the identifiers for each ion, and the obtained ion intensities of each peak were used to generate a three dimensional matrix containing arbitrarily assigned peak indices (retention time-m/z pairs), ion intensities (variables) and sample names (observations). Exacted molecular mass data (m/z) of peaks were searched through online HMDB database and KEGG database for metabolite identification. We have now added all the details in the methods section as mentioned above.

3. Confidence of metabolite identification: The authors need to describe their procedure for identifying metabolites more accurately, including the strategies for metabolite identification.

Response: Thank you, the details for identifying metabolites are now provided in more detail, as answered to question 1 and 2. Again, sorry for missing out a large proportion of metabolite identification details.

4. Statistical analysis. The integration of metabolite signals (what signals were used?) with the methylation data need to be more accurately described.

Response: Thank you, we have originally used the relative proportion (relative abundance from untargeted metabolome, in which the concentration of each metabolite was calculated as relative value to the group mean, in other words the relative proportion in comparison to the

value from mixture of all samples). Now for the reasons mentioned above and also to your additional comments, this section has been removed.

5. Functional validation. The authors use a methionine feeding study that shows that methionine changes methylation (Figure 5E). It is, however, concerning to this reviewer that the control is very similar to the high methionine containing diet. I feel this experiment would benefit from some extra controls that show that really methylation is increased by methionine, and not just the composition of the microbiome. I was not able to conclude that from the figure and the Supplemental Figure 10-12 provided.

Response: We thank the reviewer for this comment. We interpreted that the normal diet (control) already contained sufficient amount of methionine, and adding methionine in water reaches the plateau of effects in terms of methylation, thus the increase is smaller (compared to the significant decrease of gut microbial methylation in methionine-depleted diets). Nonetheless, while our experiment supports that methionine plays a causal role in methylation, we removed this section due to reasons mentioned in response to reviewer 1 and also mentioned below.

Additional comments:

6. The authors should describe in how far their procedure here is an advance because many papers have already described the use of ONT to identify and sequence the microbiome. In addition, methylation has been shown to be monitored before other groups. For instance, Tourancheau et al (Nature Methods 2021), a paper cited, succeeded in assigning methylation type 4mC, 5mC or 6mA using ONT sequencing as well. Here, only 6mA (Figure 3) is considered for further analyses.

Response: We thank the reviewer for this insight. With several ONT sequencing applied to microbiome research, so far the main focus has been assembly more complete genomes, in which >200G of reads were used for small amount of samples and not yet feasible for population studies¹ a cohort study with a focus on structural variations as well as associations with metabolic implications is first to be reported by our study; and compared to short reads, as we have discussed in detail in the manuscript, we are able to expand SVs to include also insertions, rather than previously reported studies that can only infer deletions. We have added more discussion at line 298-303.

Reference:

1. Bertrand, D. et al. Hybrid metagenomic assembly enables high-resolution analysis of resistance determinants and mobile elements in human microbiomes. *Nat Biotechnol* 37, 937-944 (2019).

For the methylation part: it was in our benchmarking process already obvious that, only part of the 6mA can be reliably detected by available approach de novo; and as suggested by reviewer 1 (see our response), currently it is not feasible to generate MDA amplified reads for all our samples, and use as reference to perform full-scale methylation analysis, we have decided to remove this section until further methodological improvement.

7. In Figure 3B, C it is unclear what the color codes refer to. In Figure 3B, the color code is too small to discriminate. While the presentation as a rose diagram in Figure 3C is fancy, I think it is not very useful and a more simple presentation (i.e. as a bar chart) will be more effective.

Response: Thank you for your valuable comments, we have modified Figure 3B, C accordingly. Specifically, the dots of the indicated colors have been enlarged and detailed in the legend.

8. Figure 3D-I Correlations may be spurious and p-values should be corrected for multiple testing.

Response: Many thanks, we have performed the Benjamini-Hochberg multiple testing in the steps of discovering associations between bacterial species and metabolites, as well as SVs and metabolites; and now noted it in Figure 3 legends (line 423-424) as well as in Results (line 218-222).

9. Table S2: Fasting blood sugar level needs a unit.

Response: Apologize, we have added in Supplementary Table 2.

REVIEWERS' COMMENTS

Reviewer #1 (Remarks to the Author):

The authors have addressed my previous comments.

Reviewer #2 (Remarks to the Author):

The authors address adequately all the comments made by reviewers, removing the section corresponding to the methylome analysis and adding sufficient additional information and analyses that greatly improve the quality of the research manuscript. After addressing the minor comments described below, I consider that this manuscript is fit for publication.

Minor comments:

The new paragraph added (lines 300-305) explains the novelty of the current approach regarding the detailed analysis of structural variation and their metabolic implications. However, it should be rewritten to improve clarity. Moreover, it repeats some information described in lines 306-308.

Supplementary Figure 2 should have a more detailed explanation in the footnote.

In Supplementary Figure 8, some of the comments added are still not clear: "(genes of cyclomalto-dextrinase (K01208), the enzyme 3.2.1.54 in purple and the enzyme 3.2.1.133 in purple is its synonym, and beta-glucosidase (K05349), the enzyme 3.2.1.21 in purple)". Rephrase this part.

Line 236: substitute "significant correlated" by "significantly correlated".

Line 237: substitute "presence of SVat ..." by "presence of SV at ..."

Line 314: remove "While"

Reviewer #3 (Remarks to the Author):

The authors have done a good job to revise the paper.

One concern that emerged in the reviewer's response is that they used retention times but did not use MS/MS data or authentic standards to identify their metabolites. This may convey problems since the structural identifications do not seem to be very confident now (see Alseekh et al, Nat Methods. 2021 Jul;18(7):747-756.).

This is a limitation that may have important ramifications for the data quality, and should be acknowledged. Alternatively, it would be useful to report all metabolites with retention times and exact masses and MS2 fragments to increase transparency.

The title might be more clear.

REVIEWERS' COMMENTS

Reviewer #1 (Remarks to the Author):

The authors have addressed my previous comments.

Response: We thank the reviewer for the input and appreciation of our study.

Reviewer #2 (Remarks to the Author):

The authors address adequately all the comments made by reviewers, removing the section corresponding to the methylome analysis and adding sufficient additional information and analyses that greatly improve the quality of the research manuscript. After addressing the minor comments described below, I consider that this manuscript is fit for publication.

Response: We thank the reviewer for the input and recommendation for publication.

Minor comments:

The new paragraph added (lines 300-305) explains the novelty of the current approach regarding the detailed analysis of structural variation and their metabolic implications. However, it should be rewritten to improve clarity. Moreover, it repeats some information described in lines 306-308.

Response: We thank the reviewer for this suggestion, it has been re-written to improve clarity and removed redundant information (line 297-300).

“With several studies have applied ONT sequencing to microbiome research, the major focus has been on improving assembly and required >200G of reads per sample, an amount not yet feasible for population studies; a cohort study with a focus on structural variations as well as associations with metabolic implications is first to be reported by our study.”

Supplementary Figure 2 should have a more detailed explanation in the footnote.

Response: Thanks, we have added some detailed explanation in Supplementary Figure 2.

In Supplementary Figure 8, some of the comments added are still not clear: “(genes of cyclomaltodextrinase (K01208), the enzyme 3.2.1.54 in purple and the enzyme 3.2.1.133 in purple is its synonym, and beta-glucosidase (K05349), the enzyme 3.2.1.21 in purple) “. Rephrase this part.

Response: Many thanks for pointing out this. We have now revised the legend of Supplementary Figure 8 to to make it more clear.

Line 236: substitute “significant correlated” by “significantly correlated”.

Response: We have now modified this grammatical error in the text (line 234).

Line 237: substitute “presence of SVat ...” by “presence of SV at ...”

Response: We have now modified this error in the manuscript (line 235).

Line 314: remove “While”

Response: Thanks, we have removed it.

Reviewer #3 (Remarks to the Author):

The authors have done a good job to revise the paper.

1. One concern that emerged in the reviewer's response is that they used retention times but did not use MS/MS data or authentic standards to identify their metabolites. This may convey problems since the structural identifications do not seem to be very confident now (see Alseekh et al, Nat Methods. 2021 Jul;18(7):747-756.).

Response: We thank the reviewer’s insights and now have a chance to present it more accurately. In the revision, we have updated the details regarding metabolites identification of metabolome analysis, and as reviewer suggested we indeed used MS/MS data for identifying metabolites. Specifically, we have now added “Exact molecular mass data (m/z) of peaks were searched through online HMDB database and KEGG database for metabolite identification. If a mass difference between observed and theoretical mass was < 10 ppm, the metabolite was annotated and the molecular formulas of the matched metabolites were further identified and validated by isotopic distribution measurements. Commercial reference standards were used to validate and confirm metabolites by comparing their MS/ MS spectra and retention time.” in the methods section (lines 437-442).

2. This is a limitation that may have important ramifications for the data quality, and should be acknowledged. Alternatively, it would be useful to report all metabolites with retention times and exact masses and MS2 fragments to increase transparency.

Response: We appreciate the reviewer’s comment and have revised the manuscript accordingly. The details for identifying metabolites are now provided in more detail, as answered to question 1. In addition, we have added the required details referred to Alseekh et al in the Supplementary Table 9.

3. The title might be clearer.

Response: Thanks, we have revised the title according the suggestion of editor.